# MLLM-CompBench: A Comparative Reasoning Benchmark for Multimodal LLMs

**Jihyung Kil**[*]   **Zheda Mai**[*]   **Justin Lee**   **Arpita Chowdhury**   **Zihe Wang**
**Kerrie Cheng**   **Lemeng Wang**   **Ye Liu**   **Wei-Lun Chao**
The Ohio State University
https://compbench.github.io

## Abstract

The ability to compare objects, scenes, or situations is crucial for effective decision-making and problem-solving in everyday life. For instance, comparing the freshness of apples enables better choices during grocery shopping, while comparing sofa designs helps optimize the aesthetics of our living space. Despite its significance, the comparative capability is largely unexplored in artificial general intelligence (AGI). In this paper, we introduce MLLM-CompBench, a benchmark designed to evaluate the comparative reasoning capability of multimodal large language models (MLLMs). MLLM-CompBench mines and pairs images through visually oriented questions covering eight dimensions of relative comparison: visual attribute, existence, state, emotion, temporality, spatiality, quantity, and quality. We curate a collection of around 40K image pairs using metadata from diverse vision datasets and CLIP similarity scores. These image pairs span a broad array of visual domains, including animals, fashion, sports, and both outdoor and indoor scenes. The questions are carefully crafted to discern relative characteristics between two images and are labeled by human annotators for accuracy and relevance. We use MLLM-CompBench to evaluate recent MLLMs, including GPT-4V(ision), Gemini-Pro, and LLaVA-1.6. Our results reveal notable shortcomings in their comparative abilities. We believe MLLM-CompBench not only sheds light on these limitations but also establishes a solid foundation for future enhancements in the comparative capability of MLLMs.

## 1 Introduction

The concept of "relativity" is integral in our daily lives. For example, relative freshness affects our decision to purchase fruits; relative spaciousness affects our decision to choose living or working space; relative crowdedness indicates which paths to select; (relative) change between two scenes reveals what happened to the environment. In short, the ability to compare objects, scenes, or situations and reason about their relativity is vital for us to make informed decisions, solve problems effectively, and acquire knowledge efficiently, enabling us to make sense of the surrounding world.

The recent advance of multimodal large language models (MLLMs), a.k.a. large multimodal models (LMMs), [1, 3, 58, 33, 32, 14, 6] has demonstrated promising progress toward artificial general intelligence (AGI) [65, 36] and achieved unprecedented results in a variety of vision and language (V&L) tasks, ranging from free-formed visual recognition [15, 10, 13] and visual captioning [10, 2] to visual question answering [21, 22, 53]. Yet, much less attention has been paid to tasks that involve relativity and comparison between multiple visual inputs, e.g., two images. In essence, most of the existing datasets for visual recognition [15, 10, 13] and V&L tasks [21, 2, 40, 31, 16, 65] comprise

---

[*]Equal contribution. {kil.5, mai.145}@osu.edu

38th Conference on Neural Information Processing Systems (NeurIPS 2024) Track on Datasets and Benchmarks.

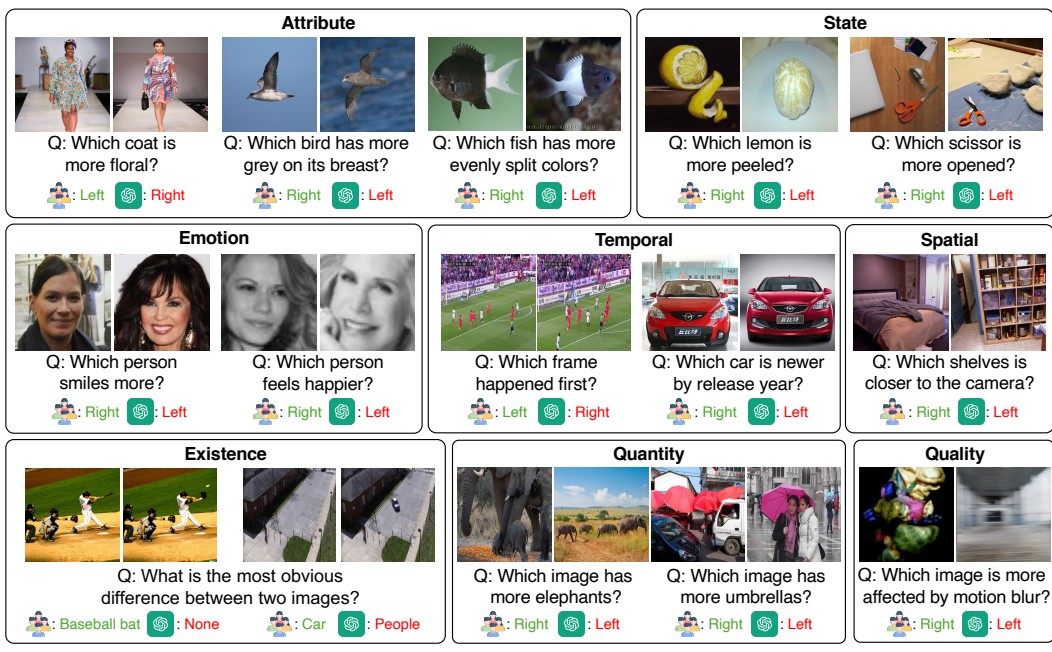

Figure 1: **MLLM-COMPBENCH** offers diverse triplets comprising two images, a question about their relativity, and an answer to cover eight types of relativity (see §1). See examples along with predictions of GPT-4V [1].

examples with only single visual inputs (e.g., an image or a video clip), making them infeasible to assess MLLMs' comparative capability.

In this paper, we introduce MLLM-COMPBENCH, a V&L benchmark dedicated to evaluating the comparative reasoning capabilities of MLLMs (Figure 1). MLLM-COMPBENCH comprises 39.8K triplets, each containing 1) a *pair* of visually or semantically relevant images 2) a question about their relativity, and 3) a ground-truth answer. We consider a wide range of questions categorized into eight aspects of relativity. **Attribute Relativity** tests the ability to recognize relative attributes [44] such as size, color, texture, shape, and pattern. For instance, given two images of birds, we ask MLLMs to compare the length of their beaks (e.g., "Which bird has longer beaks?"). **Existential Relativity** assesses the comprehension of existence in comparisons, asking questions like "Which trait is in the left butterfly but not in the right butterfly?". **State/Emotion Relativity** examines if MLLMs can identify state variations, such as different degrees of baking and smiling. **Temporal Relativity** evaluates the understanding of time-related changes between two objects or scenes (e.g., "Which video frame happens earlier during a free kick?"). **Spatial Relativity** checks the ability to tell spatial differences (e.g., "Which cup looks further?"). Finally, **Quantitiy/Quality Relativity** investigates whether an MLLM understands the relativity of quantity and quality (e.g., "Which image contains more animal instances?").

We systematically benchmark representative MLLMs on MLLM-COMPBENCH, including GPT-4V [1], Gemini1.0-Pro [58], LLaVA-1.6 [33], and VILA-1.5 [32]. Specifically, we concatenate two images horizontally (i.e., left and right) as the visual input. We then prompt MLLMs to answer questions about the relativity between these two images. When applicable, we also investigate a two-stage reasoning strategy, starting by asking a refined question about each image independently (e.g., "How many animal instances are in the image?"), followed by a pure language question (e.g., "Based on the descriptions, which image has more animal instances?"). Our results reveal notable shortcomings in existing MLLMs' comparative abilities, especially in Existence, Spatiality, and Quantity Relativity. We conduct further analyses of error cases, offering insights for future MLLMs' improvements.

In sum, MLLM-COMPBENCH has several advantages: (i) MLLM-COMPBENCH introduces new perspectives to evaluate MLLMs — comparative reasoning capabilities about relativity. (ii) MLLM-COMPBENCH provides extensive coverage across eight relativities and fourteen domains. (iii) MLLM-COMPBENCH benchmarks recent MLLMs, accompanied by detailed analyses and insights for future improvement. (iv) MLLM-COMPBENCH is extensible — we identify multiple data sources that can be further incorporated.

**Remark.** During the conference, we noticed a concurrent work by Kazemi et al. [28] that also studies MLLM's reasoning capability with multiple images, specifically focusing on math, physics, logic, code, table/chart understanding, spatial and temporal domains. We encourage readers to consult their work as well.

## 2  Related Work

**Multimodal LLMs (MLLMs).** Large Language Models (LLMs) [1, 58, 4, 5, 23, 59, 63] have made significant strides in various NLP and AI tasks. Many recent works [1, 3, 58, 33, 32, 14, 6, 29, 69, 45, 61] have extended LLMs' capabilities into the multimodal domain, particularly for vision and language (V&L) tasks. At a higher level, this advancement involves integrating a pre-trained vision encoder (e.g., CLIP [49]) with LLMs via a bridge module (e.g., an adaptor [33, 14]). Different strategies are developed to pre-train these multimodal LLMs (MLLMs), such as optimizing the LLMs and bridge module while keeping the vision encoder frozen [33] or training the bridge part only [14].

**MLLM benchmarks.** Earlier, MLLMs were evaluated on traditional V&L tasks, such as visual question answering (VQA) [21, 22, 53], image captioning [10, 2], and image-text retreival [31, 11]. Recently, a range of new and intriguing V&L tasks [37, 56] have emerged to assess MLLMs' capabilities across various dimensions. These include comprehension and reasoning about charts [38], diagrams [39], scene text [54, 40], web navigation [16], expert-level multimodal understanding [65], etc. Our MLLM-COMPBENCH complements these efforts by focusing on a new dimension, MLLMs' comparative reasoning capacity on a pair of visually or semantically relevant images.

**Multi-image datasets.** Several existing datasets [44, 57, 17, 25, 67, 27] provide multi-image data (e.g., pairs of images), but they serve different purposes (e.g., not for evaluating MLLMs) or have relatively limited scopes. NLVR2 [57] labels each image pair with a caption that may or may not be relevant to the images, asking models to predict the caption's relevance (i.e., image-text matching). A few datasets [27, 7, 66] synthesize multi-image data for instruction tuning (e.g., image editing). More relevant to ours are [17, 25, 67, 44]. Birds-to-Words [17] aims to describe the difference between two birds; Sopt-the-diff [25] focuses on the difference between two outdoor scenes; Q-bench2 [67] compares the quality (e.g., blurriness) between two images; Relative Attributes [44] compares the relativeness of attributes between two facial or natural images. However, these datasets have limited scopes, only targeting specific domains or questions. In contrast, our MLLM-COMPBENCH defines eight relative comparisons, covering a wide range of relativities in the real world. Our image pairs are curated from fourteen diverse visual domains. We believe this offers the V&L community a more comprehensive benchmark to assess the comparative capabilities of current leading MLLMs.

**Learning to rank & learning with preference.** Several research topics are relevant to ours and may benefit from our MLLM-COMPBENCH. Learning to rank (LTR) [30, 34, 8] aims to realize a scoring function that can rank examples (e.g., images) based on certain aspects, such as facial ages [41, 9] and degrees of attributes' presence [44]. Typically, an LTR model takes one example as input; the model is trained with pairs of examples such that the output scores match the ground-truth orders. Recently, learning with preference information [18] has become a mainstream approach to fine-tuning LLMs for alignment [50, 12]. Unlike our focus, these works usually collect pairs of outputs (e.g., answers to a question) with humans' preferences to supervise model fine-tuning.

## 3  Why Do We Study Comparative Reasoning?

To date, most of the existing visual recognition and V&L benchmarks focus on a single visual input (e.g., an image or a video clip), aiming to assess and promote *absolute* inference and reasoning within it, for example, identifying objects, recognizing their properties/states/actions, and describing and reasoning about their interactions within in the scene.

In reality, not all the inference and reasoning could be made absolute, or need to be absolute. For example, it is hard and ambiguous to describe the absolute degree of smiling [44], but it is relatively easy to compare two faces and tell which one smiles more. This fact applies to other visual properties like attributes (e.g., length), states (e.g., steps in cooking), and spatial locations (e.g., longitude and latitude). Often, comprehending the *relativity* is sufficient for us to make sense of the real world.

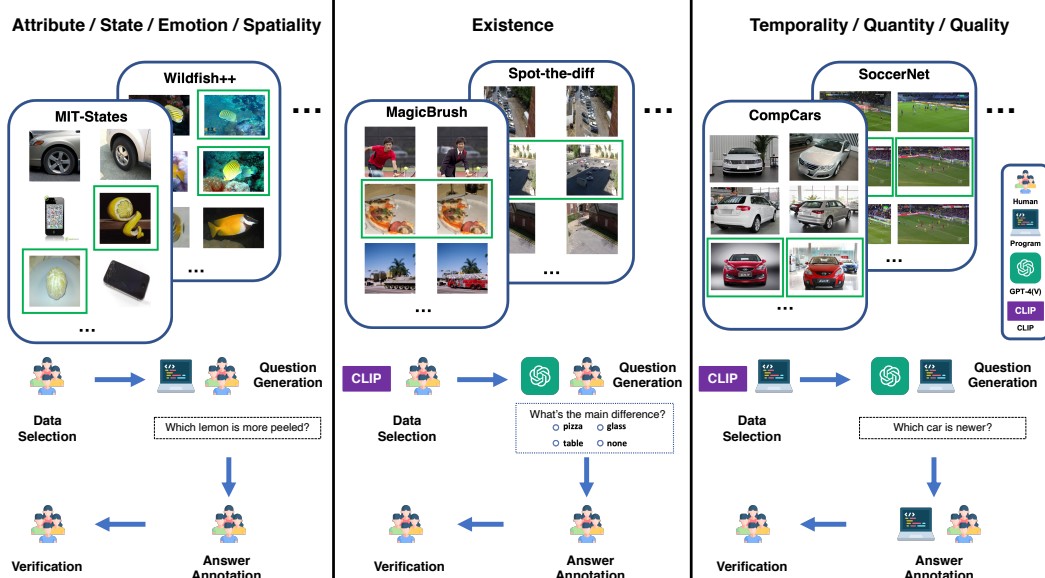

Figure 2: **MLLM-COMPBENCH curation pipeline**, including data selection, question generation, answer annotation, and verification. We rely on combinations of humans, computer programs, MLLMs (specifically GPT-4V [1]), and CLIP similarity [49] to select images and generate questions, based on relativity types and available metadata.

Furthermore, learning to infer and reason about *relativity* could naturally and more efficiently facilitate AI models to grasp *fine-grained* details. For instance, learning to describe a complex scene (e.g., captioning) often results in a model mastering common objects and properties but missing rare and subtle ones. In contrast, learning to tell the difference between two scenes promotes the model to identify subtle changes and describe them.

Last but not least, the ability to perform comparative reasoning is integral to our daily decision-making and problem-solving (see §1 for some examples). Humans' comparative capability, e.g., providing preferences between instances, has also been widely leveraged to supervise foundation models like LLMs to align their outputs with application requirements and societal expectations [50, 12]. We thus believe it is crucial to assess and promote comparative reasoning about relativity in AGI.

## 4  MLLM-COMPBENCH Benchmark

We introduce MLLM-COMPBENCH, a multimodal benchmark designed to assess the comparative reasoning abilities of MLLMs across various dimensions. In what follows, we first describe the types of comparative capabilities that MLLM-COMPBENCH aims to evaluate (§4.1). Next, we outline our methodology for collecting images, followed by how we annotate associated questions and answers to evaluate these capabilities (§4.2). Lastly, we provide detailed statistics on MLLM-COMPBENCH and discuss its data quality (§4.3). Figure 2 illustrates the overall pipeline used to develop MLLM-COMPBENCH.

### 4.1  Types of Relativity

Building upon §3, we consider eight comparison categories to evaluate MLLMs' abilities to discern differences between two similar images (Figure 1).

**(1) Visual Attribute** focuses on five common visual properties — Size, Color, Texture, Shape, and Pattern — and tests whether the model can identify the relative magnitude of these attributes between images. **(2) Existence** assesses the model's capacity to identify fine-grained variations by detecting subtle changes between images. **(3) State** involves comparing the conditions or status of objects. **(4) Emotion** assesses the model's capability to interpret degrees of human emotions. **(5) Temporality** and **(6) Spatiality** evaluate the model's ability to recognize differences in images caused by temporal or spatial differences. These categories require both commonsense and comprehension of the physical

world. Lastly, **(7) Quantity** measures the relative counting skills, and **(8) Quality** compares the quality of two images, examining the model's low-level visual perceptual skills.

## 4.2 Dataset Curation

One major challenge in constructing MLLM-COMPBENCH is mining image pairs that reflect the aforementioned relativities. Fortunately, many publicly accessible datasets in vision and V&L offer detailed annotations and metadata. We carefully investigate these datasets and identify a *seed set* of fourteen datasets that align with the eight relativity types (§4.1), covering a wide range of domains like open-domain, fashion, animal, sports, automotive, facial, and both outdoor and indoor scenes (cf. Right in Table 1). Below, we outline the datasets for each relativity type and the process for generating triplets of image pairs, a question, and an answer. *Please see the supplementary material for details.*

### 4.2.1 Visual Attribute

**Data collection.** We consider five visual attribute datasets. **MIT-States [24]** includes 245 objects with 115 visual attributes, from online sources such as food or device websites. **Fashionpedia [26]** is tailored to clothing and accessories and contains 27 types of apparel along with 294 detailed attributes. **VAW [47]**, similar to MIT-States, offers a large-scale collection of 620 unique attributes, including color, shape, and texture. **CUB-200-2011 [60]** and **Wildfish++ [47]** specifically provide attributes for birds and fish. The former catalogs 15 bird parts and their attributes (e.g., "notched tail"); the latter details 22 characteristics (e.g., "yellow pelvic fins") of various fish species. For each dataset, we cluster images by objects or parts with the same attributes (e.g., "round table", "asymmetrical blouse", "curved bill", "yellow dorsal fin") and extract visually similar image pairs from each group.

**Annotation.** We apply rule-based approaches to generate questions about relative degrees of attributes between objects (e.g., "Which coat is more floral?"). We then pair the questions with the corresponding image pairs and present them to six human annotators. The annotators are tasked with labeling the correct answers (binary: left/right) and filtering out any irrelevant or nonsensical questions about the images. In total, we construct a collection of **5.3K triplets**.

### 4.2.2 Existence

**Data collection.** We consider datasets for image editing, which provide image pairs with similar layouts but subtle changes. We adopt **MagicBrush [66]**, a recently released dataset for instruction-guided editing. It consists of (source image, instruction, target image) triplets, where the instruction specifies a subtle change between the source and target images. We also consider **Spot-the-diff [25]**, which provides image pairs in outdoor scenes, along with descriptions of their differences.

**Annotation.** We curate *multiple-choice* questions to ease automatic evaluation. We prompt GPT-4V [1] with in-context learning to generate questions; the options are formed by the extracted objects and their attributes from images. We then pass the questions (along with image pairs) to the annotators to verify the options and label the correct ones. In total, we curate **2.2K triplets**.

### 4.2.3 State

**Data collection.** We explore vision datasets covering the condition or status of objects (e.g., "pureed tomato" or "mashed potatoes"). Specifically, we use two large-scale, open-domain visual attribute datasets: **MIT-States [24]** and **VAW [47]**. They annotate not only the five common visual properties used in **Visual Attribute** but also some other properties about object states. We ask human annotators to manually review the datasets to identify image pairs relevant to state attributes.

**Annotation.** We follow the annotation protocol in §4.2.1 to curate a total of **1.1K triplets**.

### 4.2.4 Emotion

**Data collection.** We gather facial images from two publicly available datasets, **CelebA [35]** and **FER-2013 [20]**, focusing on eight annotated human emotional states: smiling, angry, disgusted, fearful, happy, neutral, sad, and surprised. We form image pairs from the same emotional state.

**Annotation.** We follow the annotation protocol in §4.2.1 to curate a total of **5.3K triplets**.

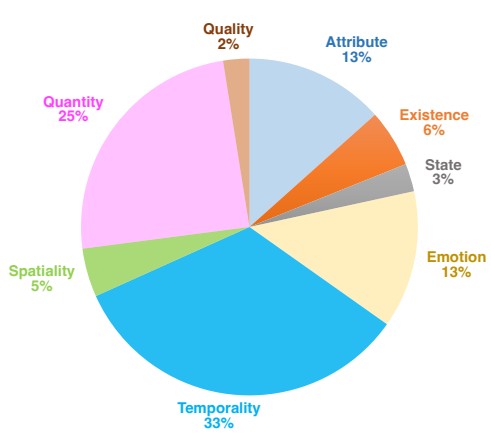

| Relativity | Dataset | Domain | # our samples |
|---|---|---|---|
| Attribute | MIT-States [24] | Open | 0.2K |
| | Fashionpedia [26] | Fashion | 2.4K |
| | VAW [47] | Open | 0.9K |
| | CUB-200-2011 [60] | Bird | 0.9K |
| | Wildfish++ [70] | Fish | 0.9K |
| Existence | MagicBrush [66] | Open | 0.9K |
| | Spot-the-diff [25] | Outdoor Scene | 1.2K |
| State | MIT-States [24] | Open | 0.6K |
| | VAW [47] | Open | 0.5K |
| Emotion | CelebA [35] | Face | 1.5K |
| | FER-2013 [20] | Face | 3.8K |
| Temporality | SoccerNet [19] | Sport | 8.3K |
| | CompCars [64] | Car | 5K |
| Spatiality | NYU-Depth V2 [55] | Indoor Scene | 1.9K |
| Quantity | VQAv2 [21] | Open | 9.8K |
| Quality | Q-Bench2 [67] | Open | 1K |
| Total | - | - | 39.8K |

Table 1: **Overall statistics of MLLM-COMPBENCH.**

### 4.2.5 Temporality

**Data collection.** We consider images with time-related tags. One pertinent source is videos. Specifically, we use **SoccerNet [19]**, a dataset for soccer video understanding. It annotates various soccer actions (e.g., free-kicks, corner-kicks, etc.) and specifies their exact periods (start-end frame indices). Using this temporal metadata, we extract two frames from each annotated action, creating an image pair that allows temporal comparison. We also consider **CompCars [64]**, a dataset designed for fine-grained categorization of vehicles. This dataset offers a detailed ontology of car attributes, such as make, model, and year. We generate image pairs that feature the same car model from different production years, for instance, a 2017 Honda Civic vs. its 2015 counterpart.

**Annotation.** We automatically generate (rule-based) questions and answers about which frame or object is associated with an earlier/later time-related tag, for example, "Which frame happened first during the free-kick?" To ensure that the two images are relevant enough to offer sufficient temporal cues, we compute the CLIP visual similarity [49], selecting only image pairs with similar layouts and object poses. In total, we curate **13.3K triplets**.

### 4.2.6 Spatiality

**Data collection.** We collect images with spatial tags, e.g., object locations. Specifically, we use **NYU-Depth V2 [55]**, featuring indoor scenes with object segments and depths. Using the segmentation maps, we identify objects within each image, and group images containing the same objects.

**Annotation.** We follow the annotation protocol in §4.2.1, leveraging pre-defined templates and object information to generate questions about spatial relative comparisons (e.g., "Which shelf is closer to the camera?"), followed by human answer annotation. Overall, we curate **1.9K triplets**.

### 4.2.7 Quantity

**Data collection.** We consider images with labels related to object instances. One prominent source is object detection datasets. Here, we use **VQAv2 [21]**, which is built upon MSCOCO [10] and encompasses a variety of question types, such as object counting and color. We focus on the counting questions, grouping images with similar questions and sampling image pairs within each group.

**Annotation.** We use GPT-4 [1] to convert original absolute counting questions (e.g., "How many elephants are there?") to relative counting questions (e.g., "Which image has more elements?"). The answers are derived automatically from VQAv2's ground-truth answers. We curate **9.8K triplets**.

### 4.2.8 Quality

**Data collection.** We use **Q-bench2 [67]**, a recently introduced dataset to evaluate low-level visual perception. Concretely, it challenges MLLMs to determine the quality (e.g., blurriness or distortion) of a single image or to compare the quality between two images.

**Annotation.** Through a meticulous filtering process (cf. §4.2.1), we select paired images from Q-bench2, along with the annotated multiple-choice questions and answers, resulting in **1K triplets**.

### 4.3 Quality Control and Dataset Statistics

To ensure the integrity of MLLM-COMPBENCH, we ask annotators to exclude poor-quality examples, such as those with low-resolution images or questions that are irrelevant or nonsensical about the images. The annotators also filter out image pairs with ambiguous relativities, for example, image pairs with indistinguishable smiling degrees. To faithfully assess fine-grained capabilities, we also apply the CLIP visual similarity to **Existence**, removing image pairs with salient differences. Additionally, we implement a rigorous cross-verification process, where each annotator confirms the accuracy of others' answers. Only samples that receive unanimous approval from annotators are kept. Consequently, our MLLM-COMPBENCH benchmark comprises **39.8K** diverse triplets (eight relativities from fourteen visual domains) with high quality and reliability. Please see Table 1 for the statistics.

**Human Annotators & Evaluators.** We recruited five in-house human annotators from our research team to work on MLLM-COMPBENCH. The annotators are instructed to avoid generating any personally identifiable information or offensive content during the annotation process. Furthermore, we recruited another five human evaluators, who were not involved in the annotation, to measure the upper bound model performance on MLLM-COMPBENCH (Table 4). The workloads for annotation and evaluation were distributed equally among annotators and evaluators.

## 5 Experiments

### 5.1 Experimental Setup

**Baselines.** We use our COMPBENCH [2] to evaluate several leading MLLMs. This includes two powerful proprietary models, GPT-4V(ision) [1] and Gemini1.0-Pro[3] [58], and two open-source alternatives, LLaVA-1.6 [33] and VILA-1.5 [32]. GPT-4V(ision) and Gemini excel in various vision and language tasks, such as VQA [21], OCR interpretation [40], spatial reasoning [38], and college-level subject knowledge [65]. LLaVA-1.6 and VILA-1.5 also demonstrate competitive performance against these proprietary giants on some tasks. Our focus is to investigate whether these cutting-edge models can extend their capabilities to the realm of multi-image relative comparison. We evaluate proprietary models via their official APIs and open-source models using (or fine-tuning on) NVIDIA RTX 6000 Ada GPUs. For more details, please refer to the Appendix C in supplementary material.

**Evaluation tasks & metrics.** We divide our COMPBENCH into a test split (31.8K) and a held-out split (7.9K), using an 80:20 ratio. The latter is reserved for future developments (e.g., prompt engineering). By default, we concatenate the image pairs horizontally (i.e., left and right) as the visual input to MLLMs, and prompt MLLMs to answer questions about the relativity between these images. To facilitate automated evaluation, we include the possible answers as options in the questions. For **Existence** and **Quality**, there are multiple options (typically more than two). For **Quantity**, there are three options: left/right/same. For other types, there are binary options: left/right. We employ the standard accuracy as our evaluation metric. A question is answered correctly if the model prediction exactly matches the ground-truth answer. Further details are included in the Appendix B.

### 5.2 Main Results (Table 2)

**Overall challenges in COMPBENCH.** We observe that current MLLMs face challenges in answering relative questions in COMPBENCH (see Table 2). All MLLMs achieve averaged accuracies over the sixteen tasks (columns) below 80%, with GPT-4V reaching the highest accuracy at 74.7%. Further, a

---

[2]We use COMPBENCH and MLLM-COMPBENCH interchangeably.

[3]Due to limited public testing quota available for Gemini1.5 during our study, we opted for Gemini1.0-Pro.

| Model | Attribute | | | | | Exist. | | State | | Emot. | | Temp. | | Spat. | Quan. | Qual. | Avg |
|---|---|---|---|---|---|---|---|---|---|---|---|---|---|---|---|---|---|
| | ST | FA | VA | CU | WF | MB | SD | ST | VA | CE | FE | SN | CC | ND | VQ | QB | |
| GPT-4V | **91.8** | **89.0** | 76.9 | 71.4 | **72.1** | **58.3** | 41.9 | **92.2** | **87.8** | 91.8 | 83.4 | **71.4** | **73.7** | 56.1 | **63.8** | **73.0** | **74.7** |
| Gemini1.0-Pro | 71.9 | 76.3 | 69.3 | 59.9 | 54.9 | 53.7 | **53.0** | 81.8 | 70.7 | 60.6 | 71.2 | 55.1 | 58.2 | 56.6 | 54.6 | 59.5 | 63.0 |
| LLaVA-1.6 | 84.9 | 72.1 | **77.7** | **72.6** | 68.7 | 26.5 | 20.7 | 89.7 | 79.3 | **96.2** | **83.5** | 51.0 | 50.2 | **67.2** | 50.1 | 64.8 | 66.0 |
| VILA-1.5 | 69.9 | 66.2 | 70.9 | 55.9 | 52.0 | 49.5 | 36.8 | 71.9 | 74.5 | 57.1 | 55.6 | 51.1 | 52.9 | 51.8 | 47.7 | 64.8 | 58.0 |
| Chance level | 50.0 | 50.0 | 50.0 | 50.0 | 50.0 | 8.6 | 9.7 | 50.0 | 50.0 | 50.0 | 50.0 | 50.0 | 50.0 | 50.0 | 33.3 | 37.4 | 43.1 |

Table 2: **Overall results on COMPBENCH test split.** We evaluate four leading MLLMs across eight relative comparisons spanning sixteen tasks. The top-performing model in each task is indicated **in bold**. ST: MIT-States [24], FA: Fashionpedia [26], VA: VAW [47], CU: CUB-200-2011 [60], WF: Wildfish++ [70], MB: MagicBrush [66], SD: Spot-the-diff [25], CE: CelebA [35], FE: FER-2013 [20], SN: SoccerNet [19], CC: CompCars [64], ND: NYU-Depth V2 [55], VQ: VQAv2 [21], QB: Q-Bench2 [67].

human evaluation study on a subset of our examples indicates that GPT-4V's performance remains notably behind human capabilities, highlighting the need for substantial improvement (Table 4).

**Superiority in State & Emotion.** State relativity is an area where MLLMs demonstrate strength. For instance, GPT-4V/LLaVA-1.6 achieve 92.2%/89.7%, respectively, on MIT-states [24] for state relativity. Similarly, they demonstrate impressive performance in emotion relativity (91.8%/96.2% on CelebA [35]). Our preliminary analysis suggests that their capacity to determine the degree of emotion (e.g., smiling) relies on specific facial features such as lip curvature or visible teeth.

**Challenges in Existence.** All MLLMs show weak performance in existence relativity tasks. We attribute this to the multiple capabilities these tasks demand, including spatial understanding and precise object recognition/comparison. For instance, when an object in the left image is moved to a different location in the right image, the models need to not only recognize the same object in the right image but also understand the relative change in its position. This necessitates both robust object recognition and accurate spatial reasoning. Given that an image can contain numerous objects, the model should have a deep understanding of how the existence of them changes between images.

**Challenges in Temporality and Spatiality.** MLLMs encounter difficulties with both temporal relativity, which requires commonsense, and spatial relativity, which demands comprehension of depth perception between objects. Specifically, for the spatial task, all MLLMs perform below 70%, and notably, both proprietary models, GPT-4V and Gemini1.0-Pro, only achieve slightly above chance levels (56.1% and 56.6%, respectively). This underscores the need for further research in improving spatial relativity to advance models towards artificial general intelligence (AGI).

**Challenges in Quantity & Quality.** We observe the mediocre performance of MLLMs in quantity relativity (e.g., GPT-4V: 63.8%, VILA-1.5: 47.7%). We attribute this to the models' weak capability in accurately counting objects in images. Similarly, MLLMs struggle with assessing image quality (e.g., 73.0% of GPT-4V's accuracy). These capabilities are crucial for making informed decisions in our daily lives (cf. §1), highlighting the need for MLLMs to improve in these aspects.

**Variability in performance across domains.** The performance of MLLMs varies in different domains. For instance, they excel at comparing visual attributes of daily objects [24] and clothing [26] while struggling with those of animals (e.g., birds [60], fish [70]). This could be due to the complexity of animal features, such as feathers, scales, or markings, which are more challenging for the model to interpret compared to simpler attributes in everyday objects.

## 5.3 Further Analyses

**Two-stage reasoning.** What if we first ask MLLMs to analyze each image in a pair separately (e.g., "How far is the table from the camera that took this photo? Return a number in feet.") and use their language responses to answer a follow-up pure language question (e.g., "Based on the responses, which object is closer to the camera?")? We evaluate this two-stage reasoning approach on three comparison tasks: Existence, Emotion, and Spatiality. We find that GPT-4V, using this two-stage reasoning, performs less effectively on all three tasks (Left in Table 3). This is likely

| Model | Exist. MB | Emot. CE | Spat. ND | | Model | Temp. SN | Quan. VQ |
|---|---|---|---|---|---|---|---|
| GPT-4V | 58.3 | 91.8 | 56.1 | | LLaVA-1.6 | 51.0 | 50.1 |
| GPT-4V$_{\text{two-stage}}$ | 45.9 | 90.3 | 36.3 | | LLaVA-1.6$_{\text{fine-tuned}}$ | 93.9 | 56.6 |

Table 3: **Left: Two-stage reasoning.** Analyzing images separately and then comparing them via a pure language question reduces performance, due to challenges in absolute inference and reasoning. **Right: Fine-tuning results.** Fine-tuned LLaVA-1.6 excels in temporal relativity but falls short in quantity, struggling with counting.

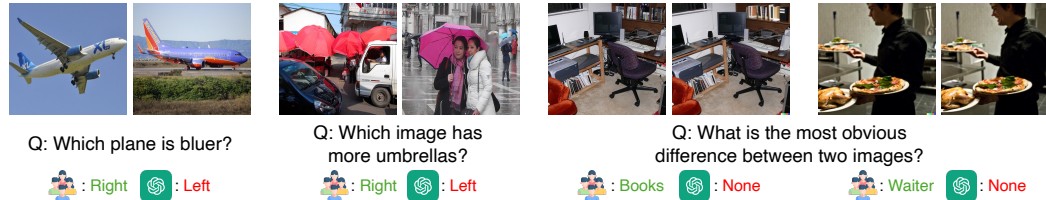

Q: Which plane is bluer? 👥: Right 🤖: Left

Q: Which image has more umbrellas? 👥: Right 🤖: Left

Q: What is the most obvious difference between two images? 👥: Books 🤖: None   👥: Waiter 🤖: None

Figure 3: **Error Analysis on COMPBENCH.** We observe four types of errors where GPT-4V [1] falls short: (i) differentiating colors between objects and backgrounds, (ii) counting small or distant objects, (iii) identifying objects within crowded scenes, and (iv) recognizing out-of-focus details.

because analyzing images separately can sometimes be more challenging than comparing images directly. For instance, calculating the exact distance from an object to the camera may be difficult, leading to inaccurate numbers. In contrast, directly answering a question, "Which object is closer to the camera?" may be easier, as models only need to determine the relative closeness between objects.

**Fine-tuning experiments.** We conduct a study to see if fine-tuning helps improve the comparative capabilities of MLLMs. We focus on two comparative tasks, temporality and quantity. For temporality, we construct a total of 20.6K training examples from SoccerNet [19], following the similar data collection and annotation protocol described in §4.2.5. For quantity, we curate 20.9K training samples from VQAv2 [21], based on the protocol in §4.2.7. We then fine-tune LLaVA-1.6 [33] on each of these training datasets separately, using LoRA techniques. As shown in Table 3 (Right), fine-tuning significantly benefits LLaVA-1.6 in the temporal task (SoccerNet). However, interestingly, it only marginal gains in quantity questions. We attribute this to its vision encoder, CLIP [49], which may have weak capabilities in counting the number of objects, as reported by several prior works [49, 43, 46]. This suggests considering new architectures or training strategies to improve its counting capabilities as future work. Please see the supplementary material for further details.

**Error Analysis.** We analyze error cases by GPT-4V and offer insights to enhance its performance (Figure 3). **First**, GPT-4V may not effectively distinguish the color between objects and backgrounds. For instance, in the first example of Figure 3, the object — a plane — shares a similar color (i.e., blue) with the background, causing GPT-4V to fail in selecting the bluer plane. **Second**, GPT-4V struggles to count accurately for small or distant objects (e.g., people further away wearing umbrellas), as shown in the second

| Model | Accuracy |
|---|---|
| GPT-4V | 68.6% |
| Humans | 86.5% |

Table 4: **Preliminary human evaluation** on 140 samples.

example. **Third**, GPT-4V finds it challenging to identify the target object if numerous items exist within images. In the third example, both images contain multiple objects, such as monitors, laptops, keyboards, desks, and books, and GPT-4V fails to pinpoint the target object (i.e., books). **Lastly**, GPT-4V may overlook details in out-of-focus areas of images. For instance, in the fourth example, the camera focuses on a pizza, leaving a waiter out of focus. Consequently, GPT-4V fails to detect facial changes in the waiter, highlighting its struggle with details in out-of-focus areas.

**Human evaluation.** We investigate how much current MLLMs (e.g., GPT-4V) lag behind human performance. We conduct a preliminary human evaluation using 140 examples randomly sampled from the sixteen tasks (columns) in Table 1. We ask five human evaluators, different from our annotators, to answer these questions and average their performance. As shown in Table 4, the performance of GPT-4V on these examples is approximately 18% below that of humans. This not

| Model | Attribute | | | | | Exist. | | State | | Emot. | | Temp. | | Spat. | Quan. | Qual. | Avg |
|---|---|---|---|---|---|---|---|---|---|---|---|---|---|---|---|---|---|
| | ST | FA | VA | CU | WF | MB | SD | ST | VA | CE | FE | SN | CC | ND | VQ | QB | |
| GPT-4V | 91.8 | 89.0 | 76.9 | 71.4 | 72.1 | 58.3 | 41.9 | 92.2 | 87.8 | 91.8 | 83.4 | 71.4 | 73.7 | 56.1 | 63.8 | 73.0 | 74.7 |
| GPT-4o | **92.3** | **97.0** | **86.3** | **74.7** | **84.5** | **81.2** | **67.2** | **95.8** | **89.6** | **96.6** | **91.1** | **72.0** | **83.3** | 68.2 | **67.8** | **81.2** | **83.1** |
| Improvement | 0.5 | 8.0 | 9.4 | 3.3 | 12.4 | 22.9 | 25.3 | 3.6 | 1.8 | 4.8 | 7.7 | 0.6 | 9.6 | 12.1 | 4.0 | 8.2 | 8.4 |
| Gemini1.0-Pro | 71.9 | 76.3 | 69.3 | 59.9 | 54.9 | 53.7 | 53.0 | 81.8 | 70.7 | 60.6 | 71.2 | 55.1 | 58.2 | 56.6 | 54.6 | 59.5 | 63.0 |
| Gemini1.5-Pro | 79.2 | 91.8 | 77.7 | 71.4 | 72.8 | 55.4 | 58.7 | 91.0 | 84.0 | 93.0 | 87.3 | 50.3 | 70.3 | **68.3** | 64.8 | 70.5 | 74.2 |
| Improvement | 7.3 | 15.5 | 8.4 | 11.5 | 17.9 | 1.7 | 5.7 | 9.2 | 13.3 | 32.4 | 16.1 | -4.8 | 12.1 | 11.7 | 10.2 | 11.0 | 11.2 |

Table 5: **Results of new MLLM models (GPT4-o and Gemini1.5-Pro) released after NeurIPS deadline.** The top-performing model in each task is indicated **in bold**. Both upgraded MLLM models (GPT-4o & Gemini1.5-Pro) exhibit significant improvements over their previous versions (GPT-4V & Gemini1.0-Pro).

only highlights the challenge of our COMPBENCH but also underscores the limited capabilities of current MLLMs in multi-image relative comparison.

## 5.4 Evaluation of Recent MLLMs Released After the NeurIPS Deadline

Since our paper submission in early June 2024, several new MLLMs have been released, such as GPT-4o [42] or Gemini1.5-Pro [52]. In Table 5, we present a comparative analysis of GPT-4V with the recently released GPT-4o, alongside Gemini1.0-Pro with Gemini1.5-Pro. These upgraded models demonstrate substantial improvements over their previous versions, with GPT-4o showing marked gains in existence (MB and SD) and spatial (ND) relativities, while Gemini1.5-Pro achieves broader enhancements across multiple relational dimensions. This progress likely results from enhanced training approaches, including scaling model and data along with refined learning strategies. Investigating exactly how these methods drive GPT-4o's substantial gains on our benchmark could be a valuable direction for future research. Nonetheless, we note that the performance of GPT-4o remains mediocre in several relativities, such as spatiality and quantity.

## 6 Conclusion and Future Work

In this work, we introduce MLLM-COMPBENCH, a comprehensive benchmark designed to evaluate comparative reasoning in multimodal LLMs (MLLMs), offering detailed analyses and insights for future advancements. As future work, we plan to incorporate more challenging datasets into each type of relative comparison in MLLM-COMPBENCH. For instance, additional video datasets could be explored for temporal relativity, such as cooking activities [68] or other sports [62, 48]. Moreover, expanding the scope of comparative reasoning relativities holds promise. Examples include similarity comparisons (e.g., "Identify similar objects between the two images.") and comparisons involving more than two images (e.g., "Given images showing various views of one object along with a few of a *different* object, the model should identify the *outliers*."). We envision that MLLM-COMPBENCH will serve as a valuable tool, paving the way for advancing comparative reasoning in MLLMs.

## Acknowledgments

This research is supported in part by grants from the National Science Foundation (IIS-2107077, OAC-2112606, OAC-2118240). We extend our sincere appreciation to our colleagues from the OSU MLB lab—Jinsu Yoo, Ping Zhang, Jiaman Wu, Ziwei Li, and Tai-Yu Pan—for their thoughtful feedback and discussions. Finally, we are grateful for the generous support of computational resources provided by the Ohio Supercomputer Center.

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

# Appendices

All codes, data, and instructions for our MLLM-COMPBENCH can be found in https://github.com/RaptorMai/CompBench. MLLM-COMPBENCH is released under a Creative Commons Attribution 4.0 License (CC BY 4.0).

Our supplementary materials are summarized as follows:

- Appendix A: Limitations, social impacts, ethical considerations, and license of assets.
- Appendix B: MLLM-COMPBENCH curation and model evaluation (cf. §4.2 and §5.1 in the main text).
- Appendix C: Training details on LLaVA-1.6 (cf. §5.3 in the main text).
- Appendix D: More qualitative examples.

## A  Discussions

### A.1  Limitations

While we conducted a human evaluation study to establish the upper bound performance on MLLM-COMPBENCH, the study is currently limited to 140 samples assessed by five evaluators (cf. §5.3 in the main text). We plan to expand the study to a larger scale in future work.

### A.2  Social impacts

MLLM-COMPBENCH evaluates the comparative reasoning abilities of MLLMs in images. A potential negative impact of our work is that malicious users might exploit our concept (i.e., comparison) to compare ethical or offensive content. Therefore, it is essential to incorporate effective safeguards in MLLMs to filter out any inappropriate materials.

### A.3  Ethical considerations

All fourteen datasets (cf. Table 1 in the main text) that we used to curate MLLM-COMPBENCH adhere to strict guidelines to exclude any harmful, unethical, or offensive content. Additionally, we instruct human annotators to avoid generating any personally identifiable information or offensive content during our annotation process. Finally, we do not conduct any study to compare harmful, ethical, or offensive content between the two images.

### A.4  License of assets

All fourteen datasets are publicly available, and Table 6 details the licensing information for the assets in each dataset. We release our MLLM-COMPBENCH under a Creative Commons Attribution 4.0 License (CC BY 4.0) to enhance global accessibility and foster innovation and collaboration in research.

## B  MLLM-COMPBENCH Curation Details

### B.1  Annotation Details

We create UI interfaces for annotation using Python in Jupyter Notebook and store the annotations in JSON files. In the following sections, we provide detailed descriptions of the annotation process for each dataset, which are omitted in the main text.

**MagicBrush** [66] is a large-scale, manually annotated dataset for instruction-guided real image editing. For each image, MagicBrush utilizes DALL-E 2 [51] to generate an edited version of the image based on language instructions, such as "let the flowers in the vase be blue." Our goal is to identify pairs of similar images. We thus use CLIP [49] to evaluate the visual similarity between the original and edited images. Only pairs exceeding a predetermined similarity threshold are selected

| Public Dataset | License |
|---|---|
| MIT-States [24] | N/A |
| Fashionpedia [26] | CC BY 4.0 |
| VAW [47] | Adobe Research License |
| CUB-200-2011 [60] | CC BY |
| Wildfish++ [70] | N/A |
| MagicBrush [66] | CC BY 4.0 |
| Spot-the-diff [25] | N/A |
| CelebA [35] | Research-only, non-commercial |
| FER-2013 [20] | N/A |
| SoccerNet [19] | MIT License |
| CompCars [64] | Research-only, non-commercial |
| NYU-Depth V2 [55] | N/A |
| VQAv2 [21] | CC BY 4.0 |
| Q-Bench2 [67] | N/A |

Table 6: **License of Assets**.

as candidate samples for our MLLM-COMPBENCH. For each selected pair, we then construct a multiple-choice question to ask the difference between two images in the pairs. Concretely, we first use GPT-4V [1] to extract all relevant objects and their attributes from the edited image with the following prompt:

> "Please extract as many components as possible from the provided images. The following examples illustrate some potential components, but the list is not exhaustive. Only provide the component names, separated by commas. If a human or an animal is shown in the images and features such as hair, eyes, hands, mouth, ears, and legs are visible, ensure to include them. Similarly, try to identify all components in as much detail as possible.
>
> Examples of components: leg, eye, ear, food, pillow, flower, plate, window, door, chair, dining table, sofa, banana, bowl, sugar, blender, berry, lizard, watermelon, motorcycle, apple, curtain, cookies, cake, hair, hat, dresses, bacon, butter, jam, bread, surfboard, t-shirt, pants, hands, fridge, plants, cabinet, sink, car, girl, boy."

We treat objects and their attributes (if found) as options for the questions. However, GPT-4V [1] may not capture all relevant objects (options) in the images. We thus request human annotators to add as many relevant options as possible. Finally, annotators are required to select the obvious difference between two images as the correct answer among options and verify the quality of the generated samples (Figure 4).

**Spot-the-diff** [25] offers video-surveillance image pairs from outdoor scenes, along with descriptions and pixel-level masks of their differences. Similar to MagicBrush, we aim to construct a multiple-choice question to find the obvious difference between the two images. We first prompt the text-only GPT-4 to extract the potentially correct objects from the descriptions of the differences using the following prompt:

> "These sentences describe the differences between the two images. Extract the objects from these sentences. for example, ["there are more people", "the car moved"], you should return "people, car". Please only provide the answer without any explanation and separate the answer names by commas."

Given the extracted objects and the images, GPT-4V is tasked with finding relevant options in the images based on the following prompt:

> "Please list all the objects and attributes associated with the image, for example, black cars, people, trees, white trucks, and yellow poles. Only provide one attribute (adjective) per object. Please only provide the answer without any explanation and separate the answer names with commas. Ensure to include these objects: [OBJECTS FROM LAST STEP]"

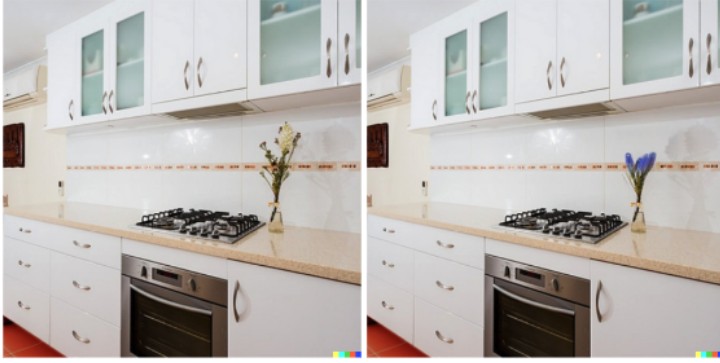

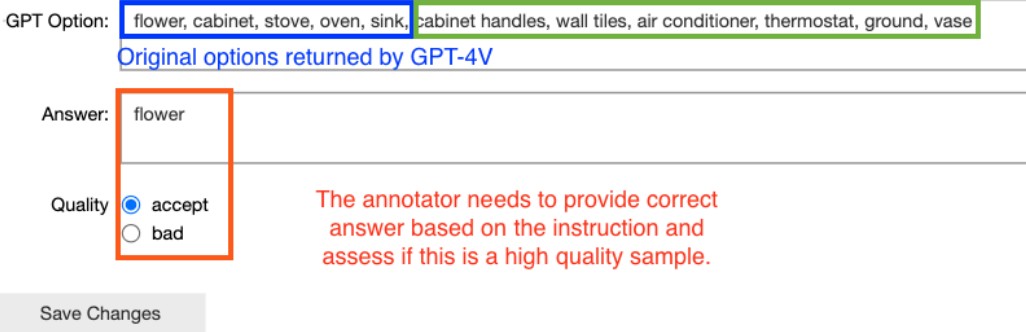

Figure 4: **Annotation Interface for MagicBrush.**

We then instruct human annotators to include additional options (if necessary) and identify the most evident difference between two images from the available options as the correct answer (Figure 5).

**MIT-States** [24] includes 245 objects with 115 visual attributes or states from online sources such as food or device websites. Each folder in this dataset is named by (adjective, noun), e.g., tall tree, where the adjective describes the state or the attributes and the noun is the object. All the images in this folder share the same adjective and noun. We apply rule-based approaches to generate questions about relative degrees of attributes or states between objects (e.g., "Which tree is taller?"). We then present the questions with the corresponding images in this folder to annotators. The annotators are tasked to select pairs from all the images, label the correct answers (binary: left/right), and filter out any irrelevant or nonsensical questions about the images. In addition, the annotators are required to determine the attribute or state types by selecting from the following options: Size, Color, Texture, Shape, Pattern, State, or None. We filter out examples where the type or answer is None. The annotation UI interface is shown in Figure 6.

**VAW** [47] provides a large-scale collection of 620 unique attributes, including color, shape, and texture. We process VAW in the same manner as MIT-States, as detailed in Figure 6.

**CUB-200-2011** [60] catalogs 15 bird parts and their attributes (e.g., "notched tail"). We group images by species with the same attributes (e.g., "curved bill") and extract visually similar image pairs from each group. We then prompt GPT-4 to transform visual attributes into questions that compare them using the following in-context prompt:

> "I want to turn some text describing the attributes of birds into a question comparing these attributes between birds in two different images. Here are some examples:
> Attribute: has_bill_shape::hooked, Questions: Which bird has a more hooked bill?

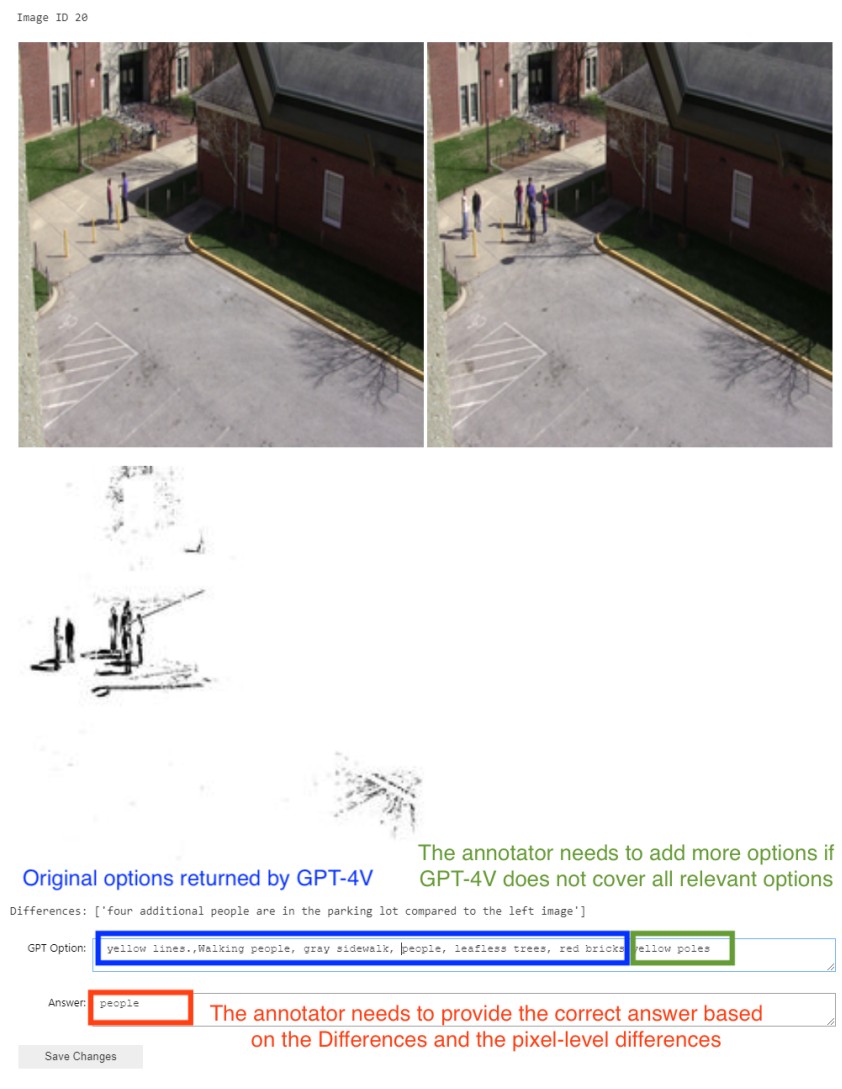

Figure 5: **Annotation Interface for Spot-the-diff.**

Attribute: has_crown_color::brown, Questions: Which bird has more brown on its crown?

Please turn this list of attributes into these questions in this format or style. I want a dictionary format output. [ATTRIBUTE LIST]"

The annotators receive all images in each group along with corresponding comparative questions generated by GPT-4. They are asked to select the pairs from the images and label the correct answers (binary: left/right). The annotation interface is shown in Figure 7.

**Wildfish++** [70] details 22 characteristics (e.g., "brown pelvic fins") of various fish species and provides detailed descriptions of the differences between two visually similar species. Using the characteristics and the descriptions of difference, we first ask annotators to generate comparative questions (e.g., "Which fish has lighter brown pelvic fins?"). Subsequently, we pass all images from the two similar species along with the corresponding question to the annotators. They select one image from each group to form a pair and label the correct answers as either left or right (Figure 8).

**Fashionpedia** [26] is tailored to clothing and accessories and contains 27 types of apparel along with 294 detailed attributes. We group images by (attribute, type), e.g., square neckline. We apply rule-based approaches to generate questions about relative degrees of attributes (e.g., "Which neckline is more square?") for each group. We then present images of the same type with different attributes,

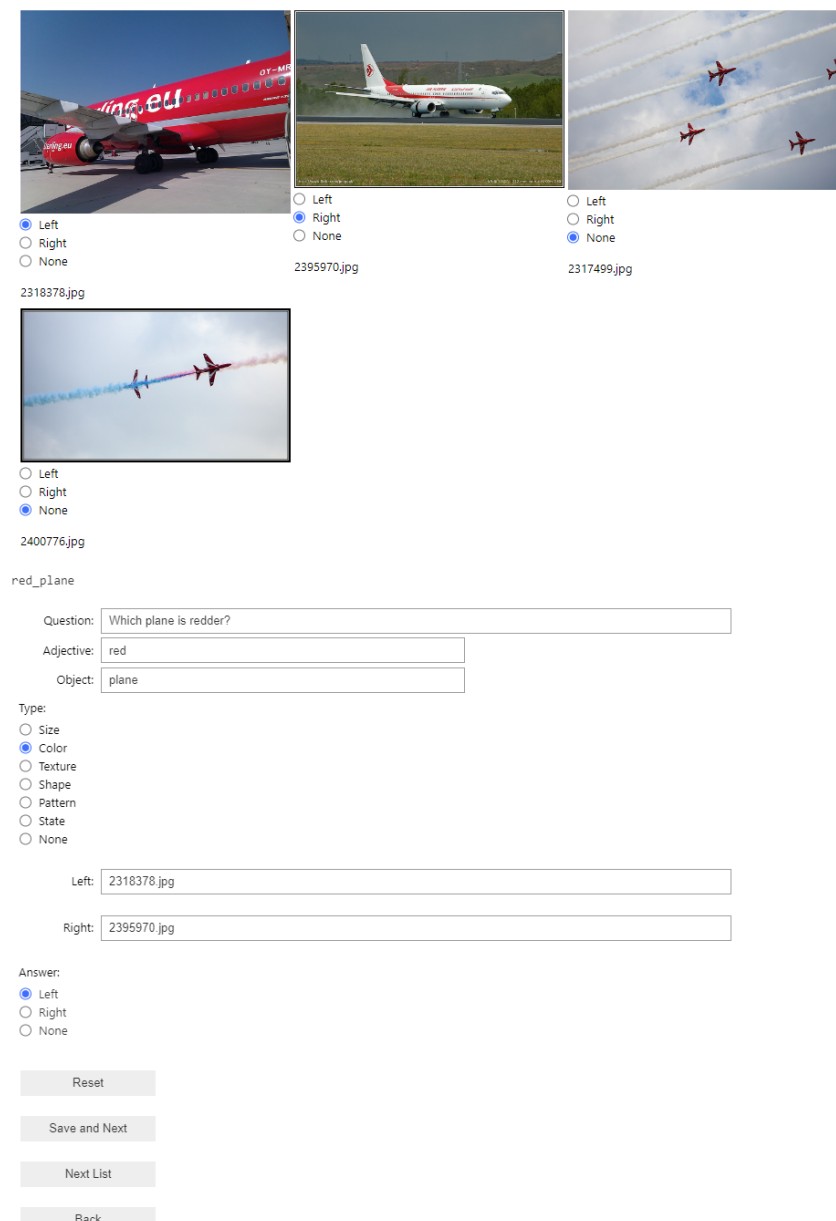

Figure 6: **Annotation Interface for MIT-States and VAW.**

such as "square neckline" and "oval neckline" to the annotators. The annotators are required to select one image from each group to form a pair, choose one between questions from two attributes, and label the correct answer (binary: left/right). The annotation UI interface is shown in Figure 9.

**NYU-Depth V2** [55] features indoor scenes with object segments and depths. Using the segmentation maps, we identify objects within each image and group images containing the same objects. We apply rule-based approaches to generate questions about spatial relative comparisons (e.g., "Which [OBJECT] is closer to the camera?"). The annotator needs to select pairs from all the images in the same group and label the correct answers either left or right (Figure 10).

**CelebA** [35] is a large-scale facial attributes dataset featuring over 200K celebrity images, each annotated with 40 attributes. We focus on images labeled with the "smiling" attribute, as it is the only attribute related to the emotion in the dataset. We generate a comparative question such as "Which

person smiles more?". The annotators are tasked with selecting pairs from all images with the smiling attribute and labeling the correct answers either left or right (Figure 11).

**FER-2013** [20] contains grayscale images along with categories describing the emotion of the person, including Angry, Disgust, Fear, Happy, Sad, Surprise, and Neutral. We leverage rule-based approaches to generate questions about relative emotional comparisons (e.g., "Which person looks more [EMOTIONAL ADJECTIVE]?"). The annotators are required to select pairs from images that share the same emotional attribute and determine the correct answers as either left or right (Figure 12).

**SoccerNet [19], CompCars [64], VQAv2 [21], Q-bench2 [67]** are automatically processed to generate samples for MLLM-COMPBENCH using their metadata and CLIP visual similarity. For more details, please refer to §4.2 of the main text.

### B.2 Language Prompts for MLLMs

Table 7 summarizes our language prompts for evaluating MLLMs. We observe that in the case of SoccerNet [19], Gemini1.0-pro [58] always predicts the answer "Left" for binary questions (e.g., "These are two frames related to [SOCCER_ACTION] in a soccer match. Which frame happens first? Please only return one option from (Left, Right) without any other words."). We thus prompted the Gemini to answer open-ended questions (as shown in Table 7) instead. We then task human evaluators with verifying whether its responses (i.e., textual descriptions) match the ground-truth answers to calculate its performance. For a fair comparison, we apply the same open-ended questions to other models (i.e., GPT-4V [1], LLaVA-1.6 [33], VILA-1.5 [32]) and report their accuracies.

### B.3 Model Evaluation

We use official APIs to evaluate proprietary MLLMs, GPT-4V [1] and Gemini [58]. For GPT-4V, we use the version of gpt-4-turbo[4]. For Gemini, we use the Gemini1.0 Pro Vision[5]. For open source models such as LLaVa-1.6-34b [33][6] and VILA-1.5-40b [32][7], we utilize their official source codes and conduct inference on NVIDIA RTX 6000 Ada GPUs.

## C Training details on LLaVA-1.6

As discussed in §5.3 of the main text, we conduct a study to evaluate whether fine-tuning enhances the comparative capabilities of MLLMs. Concretely, we focus on two relativities: Temporality and Quantity. For temporality, we construct a total of 20.6K training examples from SoccerNet [19], following the similar data collection and annotation protocol described in §4.2.5 of the main text. For quantity, we curate a total training set of 20.9K samples from VQAv2 [21], based on the similar data collection and annotation pipeline in §4.2.7 of the main text. We fine-tune LLaVA-1.6-34b [33] on each of these training datasets separately, using LoRA techniques. We follow similar hyperparameter settings as those provided in the official LLaVA source codes. For instance, batch size/the number of epochs/learning rate are 16/3/2e-5, respectively. See the training script in our GitHub repository for the complete configuration. All models are fine-tuned on four NVIDIA RTX 6000 Ada GPUs.

## D More qualitative examples

In addition to the main text, we show more qualitative examples from each of fourteen datasets in Figure 13, Figure 14, Figure 15, Figure 16, and Figure 17. We observe that GPT-4V, one of the leading MLLMs, often faces challenges across a range of relative comparison tasks.

---

[4] https://platform.openai.com/docs/models/gpt-4-turbo-and-gpt-4
[5] https://ai.google.dev/gemini-api/docs/models/gemini#gemini-1.0-pro-vision
[6] https://github.com/haotian-liu/LLaVA
[7] https://github.com/Efficient-Large-Model/VILA

| Dataset | Model | Lagnauge Prompt |
|---|---|---|
| ST, FA, VA, CU, WF, CE, FE, ND | GPT-4V LLaVA-1.6 VILA-1.5 | "[QUESTION] If you choose the first image, return Left, and if you choose the second image, return Right. Please only return either Left or Right without any other words, spaces, or punctuation." |
| | Gemini1.0-pro | "[QUESTION] If you choose the first image, return First, and if you choose the second image, return Second. Please only return either First or Second without any other words, spaces, or punctuation." |
| MB, SD | GPT-4V LLaVA-1.6 VILA-1.5 Gemini1.0-pro | "What is the most obvious difference between the two images? Choose from the following options. If there is no obvious difference, choose None. Options: None, [OPTIONS]. Please only return one of the options without any other words. " |
| SN | GPT-4V LLaVA-1.6 VILA-1.5 Gemini1.0-pro | "These are two frames related to [SOCCER_ACTION] in a soccer match. Which frame happens first?" |
| CC | GPT-4V LLaVA-1.6 VILA-1.5 | "Based on these images, which car is newer in terms of its model year or release year? Note that this question refers solely to the year each car was first introduced or manufactured, not its current condition or usage. If you choose the first image, return Left, and if you choose the second image, return Right. Please only return either Left or Right without any other words, spaces, or punctuation." |
| | Gemini1.0-pro | Based on these images, which car is newer in terms of its model year or release year? Note that this question refers solely to the year each car was first introduced or manufactured, not its current condition or usage. If you choose the first image, return First, and if you choose the second image, return Second. Please only return either First or Second without any other words, spaces, or punctuation." |
| VQ | GPT-4V LLaVA-1.6 VILA-1.5 Gemini1.0-pro | "[QUESTION] If the second image has more, return Right. If the first image has more, return Left. If both images have the same number, return Same. Please only return either Left or Right or Same without any other words, spaces, or punctuation." |
| QB | GPT-4V LLaVA-1.6 VILA-1.5 Gemini1.0-pro | "[QUESTION] Options: [OPTIONS]" |

Table 7: **Language prompts for evaluating MLLMs**. ST: MIT-States [24], FA: Fashionpedia [26], VA: VAW [47], CU: CUB-200-2011 [60], WF: Wildfish++ [70], MB: MagicBrush [66], SD: Spot-the-diff [25], CE: CelebA [35], FE: FER-2013 [20], SN: SoccerNet [19], CC: CompCars [64], ND: NYU-Depth V2 [55], VQ: VQAv2 [21], QB: Q-Bench2 [67].

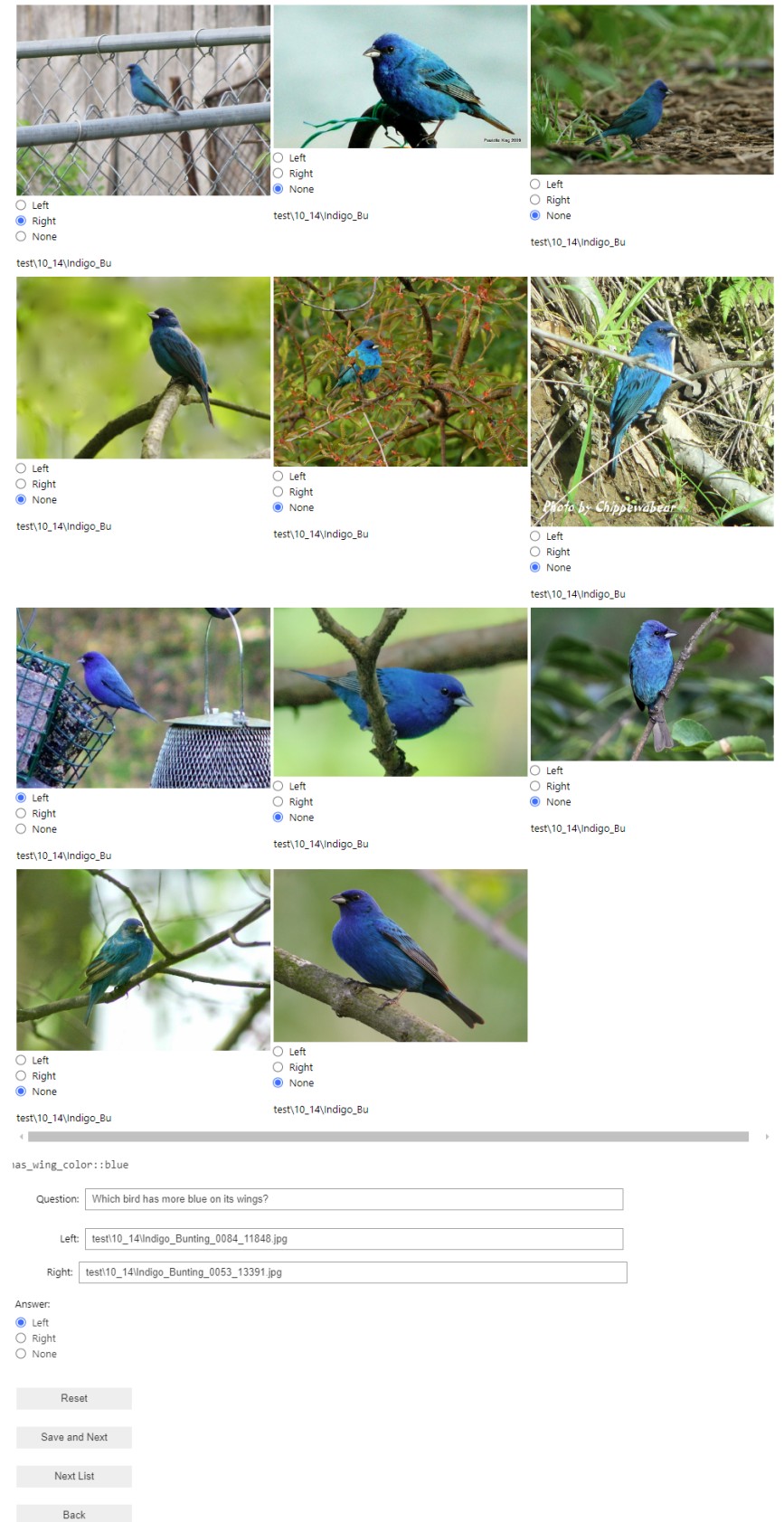

Figure 7: **Annotation Interface for CUB-200-2011.**

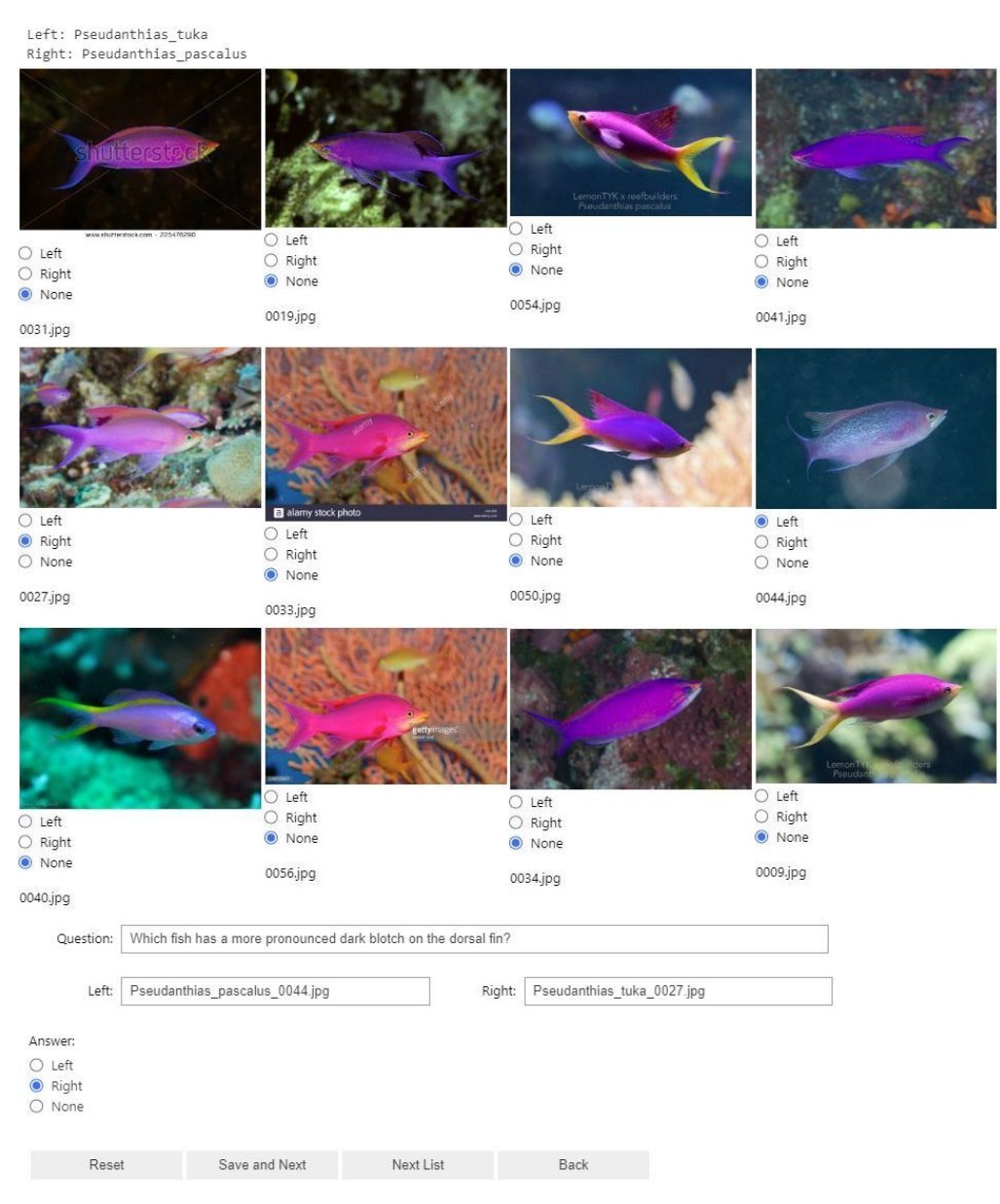

Figure 8: **Annotation Interface for Wildfish++.**

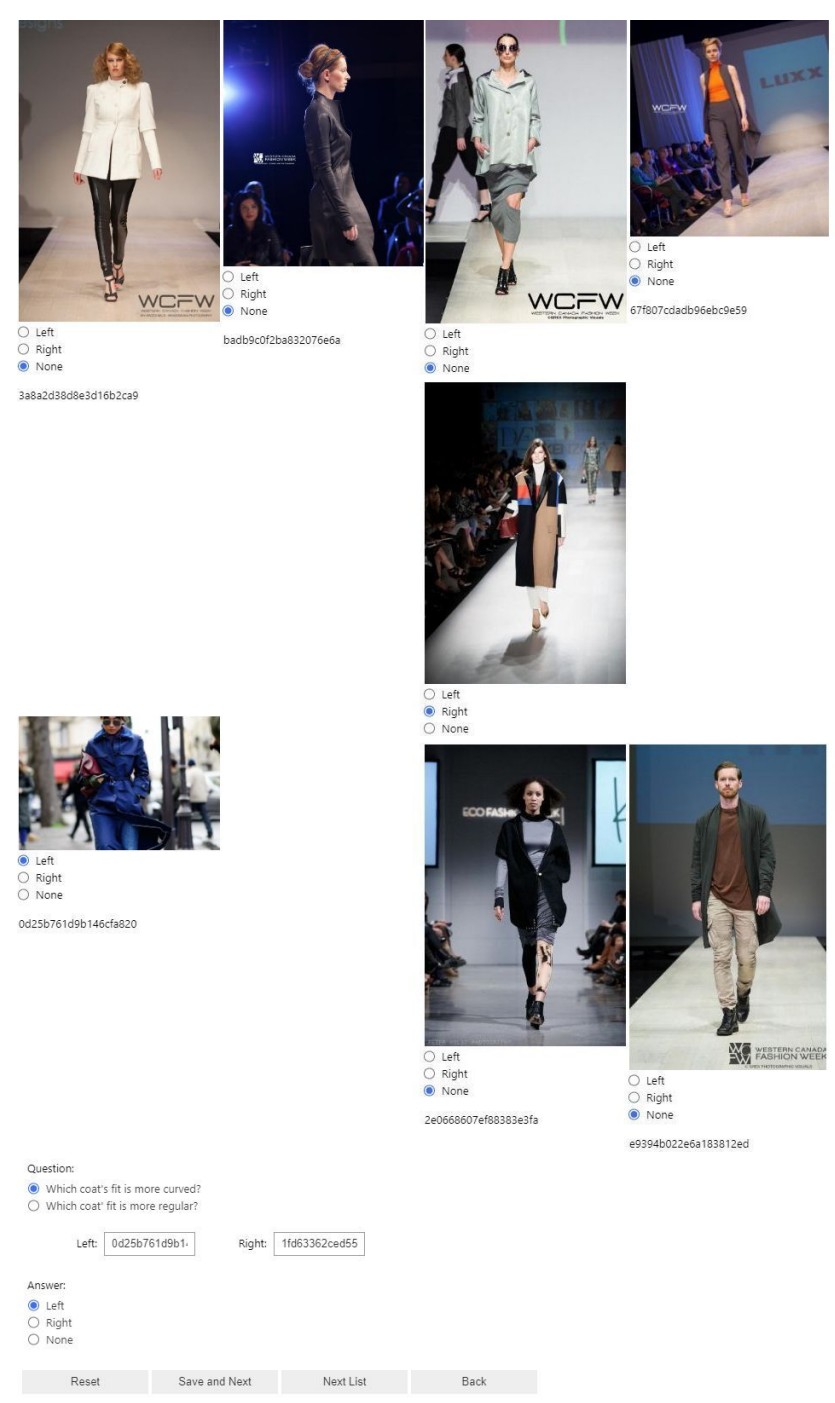

Figure 9: **Annotation Interface for Fashionpedia.**

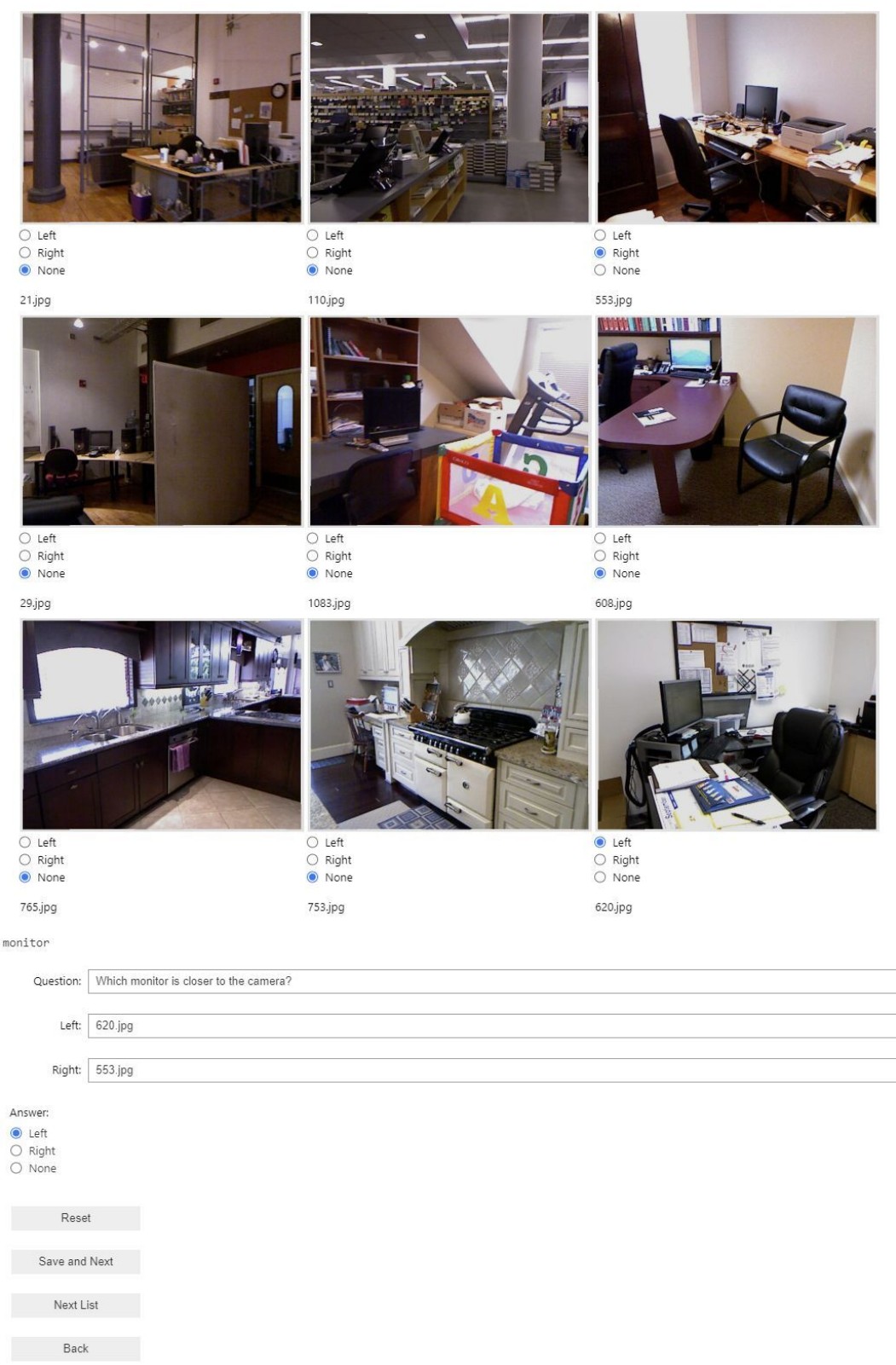

Figure 10: **Annotation Interface for NYU-Depth V2.**

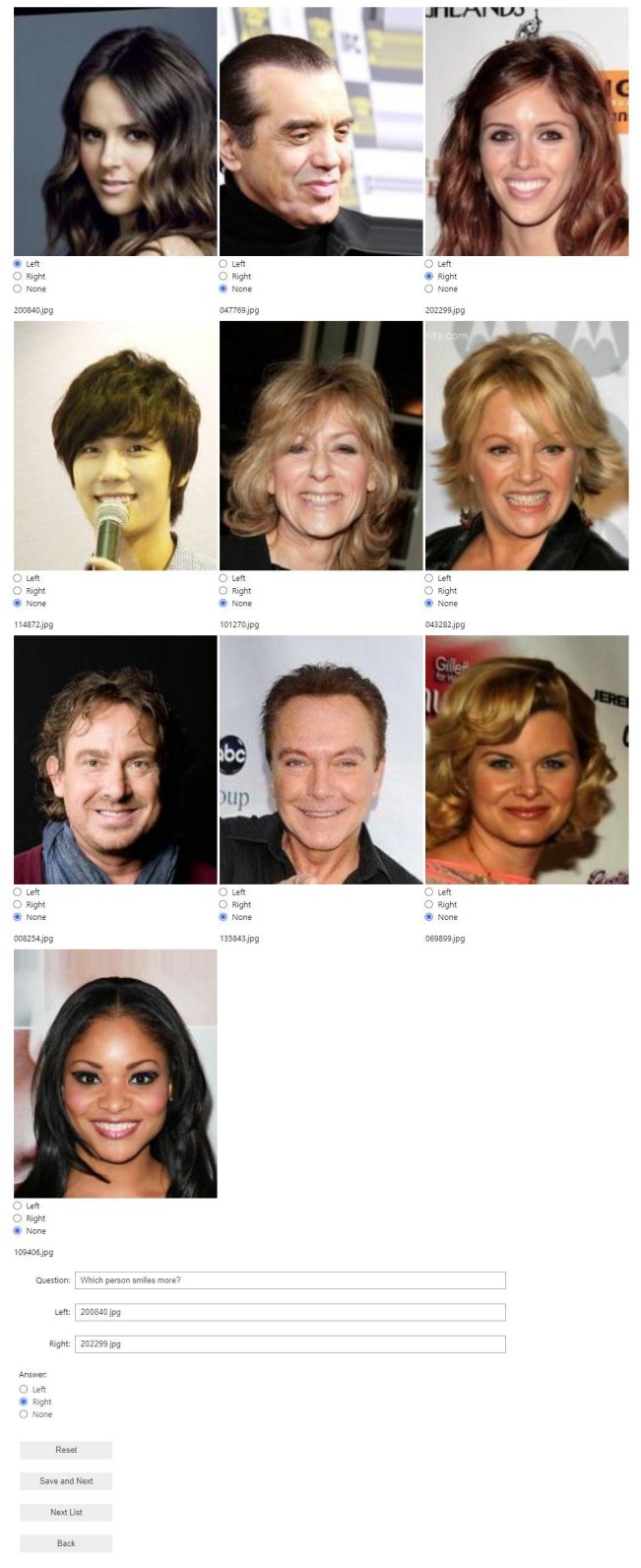

Figure 11: **Annotation Interface for CelebA.**

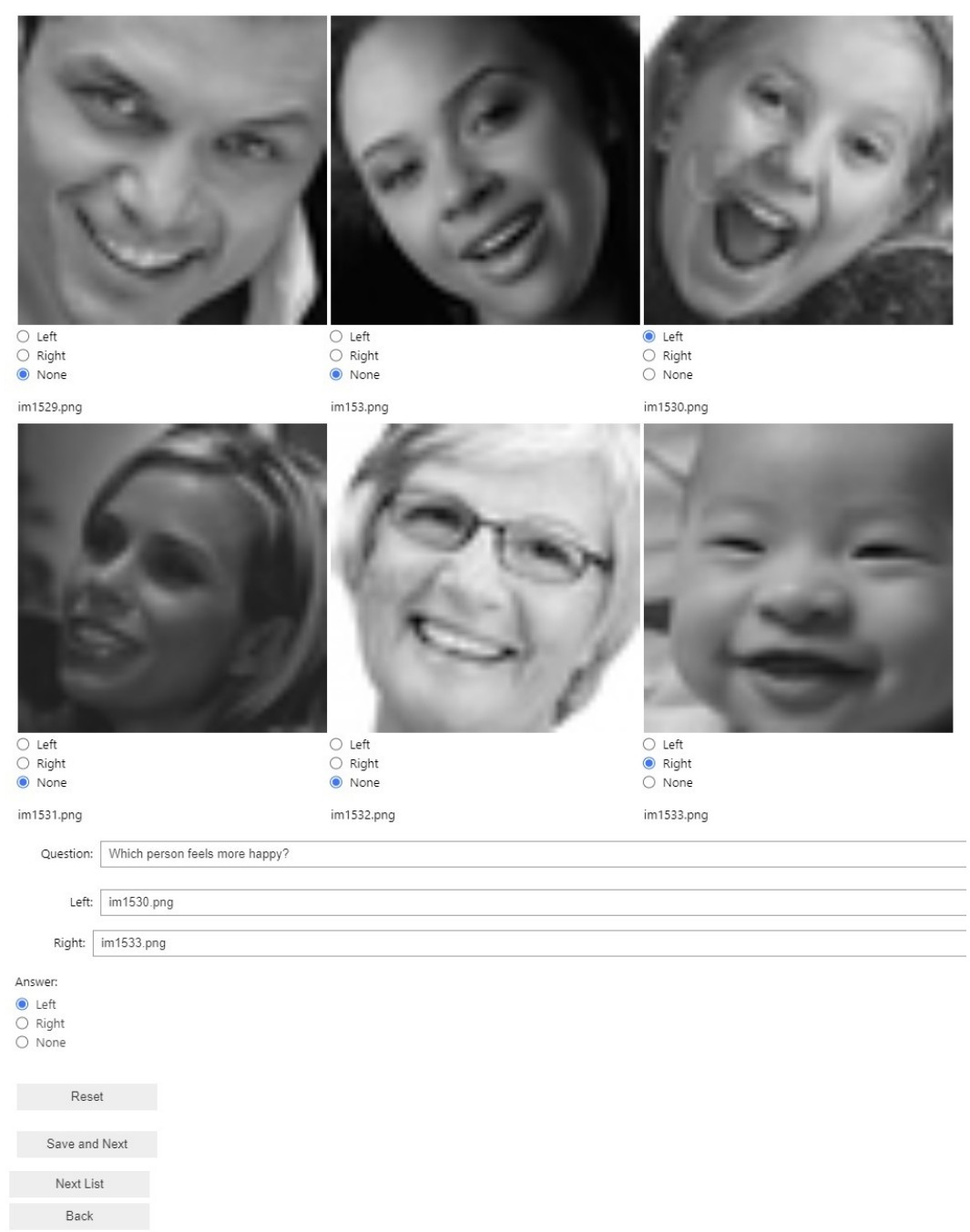

Figure 12: **Annotation Interface for FER-2013.**

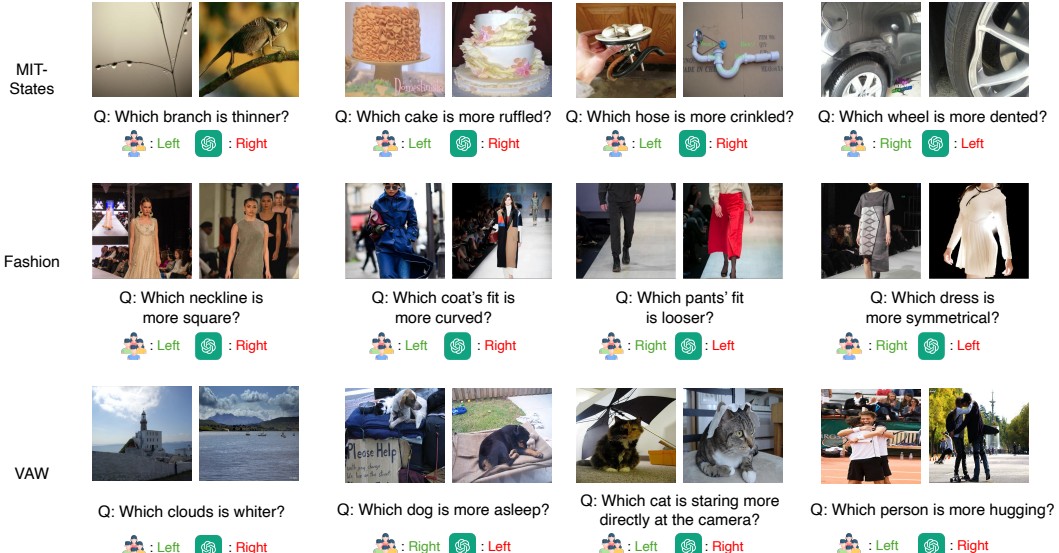

Figure 13: **Qualtiative examples on MIT-States [24], Fashionpedia [26], and VAW [47].**

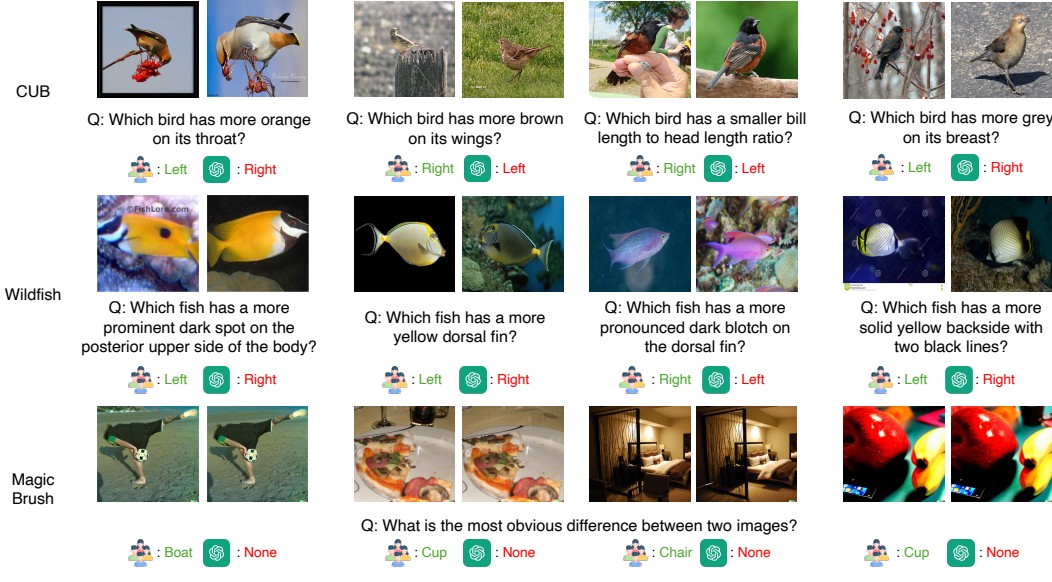

Figure 14: **Qualtiative examples on CUB-200-2011 [60], Wildfish++ [70], and MagicBrush [66].**

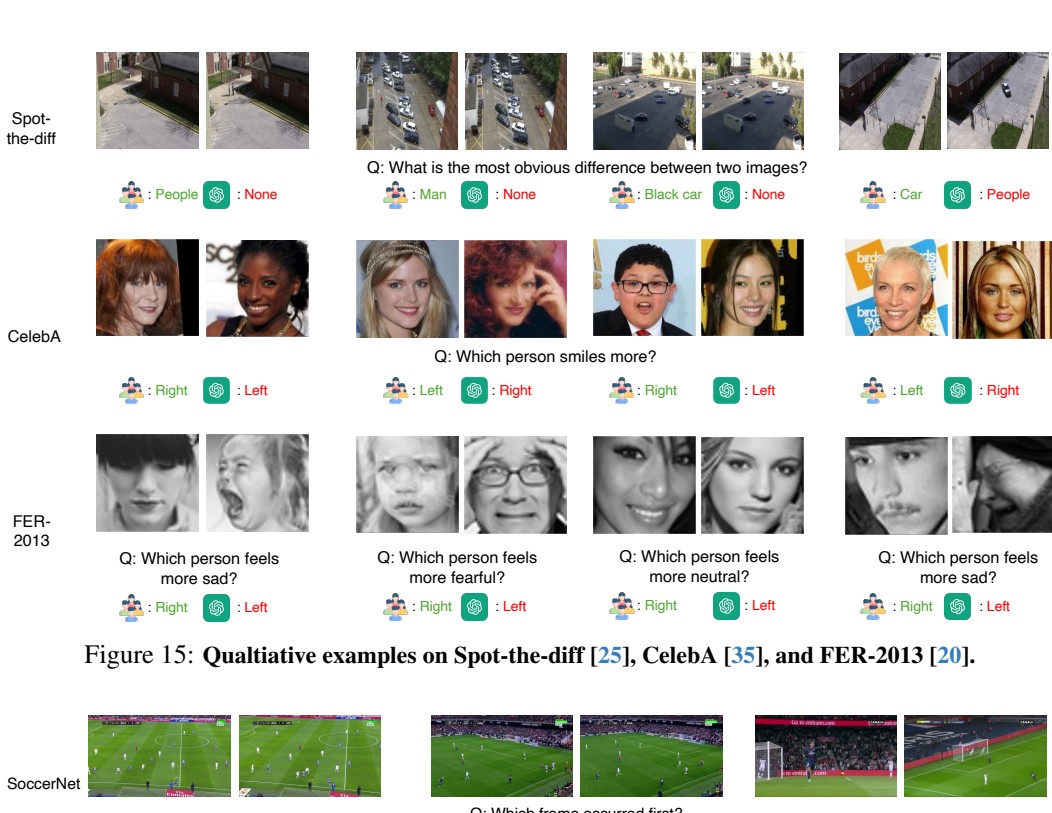

Figure 15: **Qualtiative examples on Spot-the-diff [25], CelebA [35], and FER-2013 [20].**

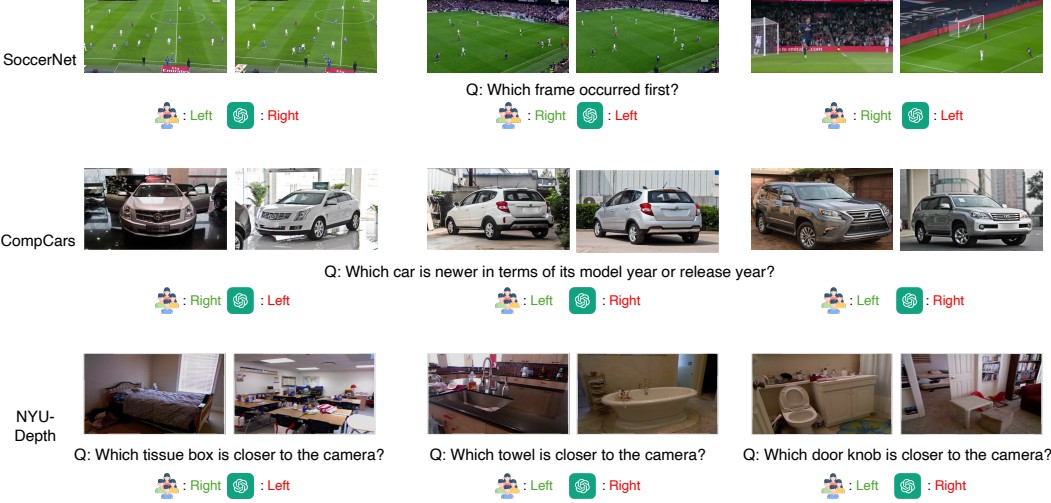

Figure 16: **Qualtiative examples on SoccerNet [19], CompCars [64], and NYU-Depth V2 [55].**

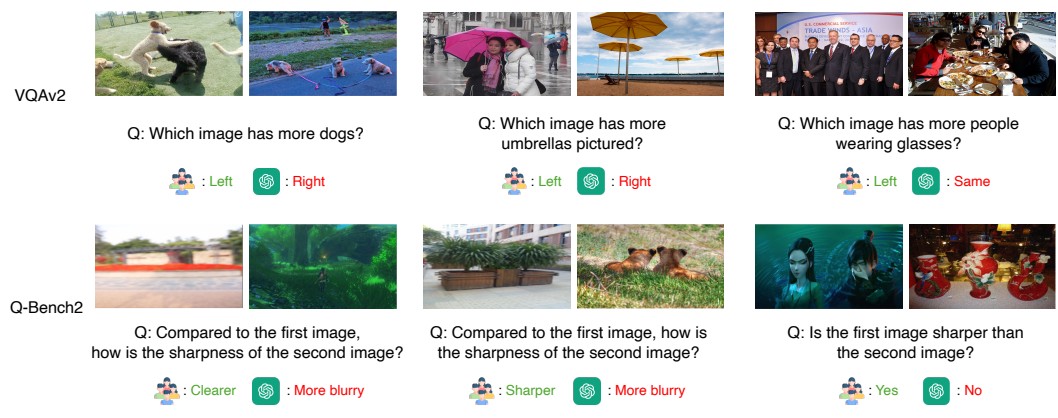

Figure 17: **Qualtiative examples on VQAv2 [21] and Q-Bench2 [67].**

