# OpenReview forum: "MLLM-CompBench: A Comparative Reasoning Benchmark for Multimodal LLMs"
_NeurIPS.cc/2024/Datasets_and_Benchmarks_Track — NeurIPS 2024 Track Datasets and Benchmarks Poster_

### Official Review · Reviewer_xgQo · 2024-07-15
**A comparative dataset for MLLMs**

**Rating:** 6
**Confidence:** 3
**Correctness:** Yes
**Clarity:** Yes

**Review:**

This paper focuses on comparative question testing for MLLMs.  COMPBENCH covers a broad spectrum of visual domains and relativity types, offering a robust platform for evaluating a wide range of comparative reasoning skills in MLLMs. The dataset is carefully curated using a combination of human annotators, MLLMs, and CLIP similarity scores, ensuring high-quality and relevant comparative examples. The main short coming of this paper is the limited scope (only focus on comparative questions). It only provides an add on to existing MLLM benchmark.

**Strengths:**

The paper provides a thorough analysis of model performances, including error cases, which can guide future research and development in improving MLLMs' comparative reasoning abilities.

**Additional Feedback:**

N/A

**Documentation:**

Yes

**Limitations:**

Yes

**Opportunities For Improvement:**

1. Expand the idea of comparative comparison to different level of difficulties, from simple appearance comparison to multi-step reasoning.
2. report the random guess performance to show the dataset is balanced.

**Relation To Prior Work:**

Yes

**Summary And Contributions:**

The paper introduces COMPBENCH, a new benchmark designed to evaluate the comparative reasoning capabilities of multimodal Large Language Models (MLLMs). The benchmark consists of around 40K image pairs across various visual domains, each paired with a question and answer that focus on eight dimensions of relative comparison: visual attribute, existence, state, emotion, temporality, spatiality, quantity, and quality. The authors curated these image pairs using metadata from existing vision datasets and CLIP similarity scores. They evaluated several recent MLLMs, including GPT-4V, Gemini-Pro, and LLaVA-1.6, using COMPBENCH and found significant shortcomings in their comparative abilities, particularly in existence, spatiality, and quantity relativity. The paper also provides a detailed analysis of error cases and insights for future improvements.

1. The creation of COMPBENCH, a comprehensive benchmark for evaluating comparative reasoning in MLLMs across eight relativity dimensions.
2. A collection of 40K annotated image pairs covering a wide range of visual domains for assessing MLLMs’ comparative capabilities.
3. Detailed evaluations and analyses of recent MLLMs, revealing their limitations and providing insights for future enhancements.

---

> ### Author Rebuttal · Authors · 2024-08-17
>
> **Q. [Limited scope (only focus on comparative questions)]** Thank you for the comment. While we are unsure whether the limited scope mentioned in your review section refers to the “weakness” of our paper, we would like to share our thoughts on this aspect. Although we focus on comparative questions, we respectfully think that such a *focused* scope is not necessarily a *limitation*. In our humble opinion, even within comparative questions, there are various types of relativity (we identified eight), indicating their inherent breadth and diversity. This variety merits a focused study to explore these questions in depth.
>
> We also respectfully think that positioning our benchmark as a “complementary” add-on to existing benchmarks is not a weakness. Indeed, when developing the benchmark, we tried to prevent ourselves from “reinventing the wheel” in areas that are already well-established. We spent significant time surveying existing benchmarks to identify what critical aspects have been missing, which led us to the comparative capability.  We will add a discussion in the final version on how our benchmark can be used alongside existing ones to provide a comprehensive evaluation of MLLMs.
>
> **Q. [Expand the idea of comparative comparison to different levels of difficulties]** This is a very good suggestion. It will create an orthogonal, complementary dimension to our benchmark: besides different relativity types, for each type, we could also separate the levels of difficulties. We will include a discussion about this potential extension in our final version.
>
> **Q. [Random guess performance & dataset balance]** Thank you for the valuable comment. In our humble opinion, our reported chance rate is equivalent to the random guess performance: randomly selecting one answer from the answer candidate pool. However, we agree that this could not reflect whether the dataset is balanced. We plan to instead report the “performance by blindly choosing the most frequent answer.” The gap between this accuracy and the chance rate would reflect the dataset imbalance.

---

> ### Comment · Reviewer_xgQo · 2024-08-29
>
> Thank you for your response. My score will stay.

---

### Official Review · Reviewer_MVH6 · 2024-07-17
**Review: CompBench: A Comparative Reasoning Benchmark for Multimodal LLMs**

**Rating:** 6
**Confidence:** 3
**Clarity:** The paper is well written.

**Review:**

The paper provides a novel focus on comparative reasoning in MLLMs, proposing a novel benchmark CompBench by curating a diverse dataset and incorporating MLLM evaluations. This work is significant for its potential to drive advancements in MLLMs, highlighting current limitations and influencing future research in AI applications requiring nuanced visual comparisons. The strengths can be summarized as following:
- Innovative Idea: The paper introduces a novel perspective, focusing on comparative reasoning, which is often overlooked in current benchmarks.
- Comprehensive Coverage: CompBench covers eight comparative dimensions and includes data from fourteen different domains and the data scale is very large.

However, there are still some weaknesses:
- Data Bias: Despite covering multiple domains, the data sources for different types of questions vary, potentially introducing some bias.
- Question Generation: The rule-based method for generating questions is relatively simple, which might limit the diversity of the questions obtained.
- Limited Baselines: The paper includes few baselines in the experiments, especially under the fine-tuning settings.

**Strengths:**

- The introduction of CompBench represents a significant advancement in the evaluation of multimodal large language models (MLLMs), specifically focusing on comparative reasoning. This fills a crucial gap in existing benchmarks, which typically assess absolute reasoning within single images or video clips.
- By covering eight distinct comparative reasoning categories, the benchmark can evaluate a wide range of comparative reasoning capabilities, making it a valuable resource for advancing MLLM development.
- By highlighting the current limitations of MLLMs in comparative reasoning, the paper provides clear directions for future research. This can inspire further innovations and improvements in MLLM architectures, training methods, and evaluation strategies.

**Additional Feedback:**

N/A.

**Correctness:**

The evaluation methods and experiment design conducted in the paper appear to be correct.

**Documentation:**

Yes.

**Ethics:**

N/A.

**Limitations:**

N/A.

**Opportunities For Improvement:**

- Address Data Bias: To reduce potential data bias, consider standardizing data sources across different types of questions or implementing techniques to balance and normalize the dataset.
- Enhance Question Generation: Improve the diversity of generated questions by employing more sophisticated methods, such as leveraging advanced LLMs or incorporating human-in-the-loop approaches for generating and validating questions.
- Expand Baselines: Include a wider range of baseline MLLMs in the experiments to provide a more comprehensive evaluation of CompBench's performance and comparative reasoning capabilities.

**Relation To Prior Work:**

The differences between this work and prior works are discussed clearly.

**Summary And Contributions:**

This paper introduces CompBench, a comprehensive multimodal benchmark designed to evaluate the comparative reasoning abilities of multimodal large language models (MLLMs), which aims to assess the models’ proficiency in discerning relative differences across various visual properties, such as size, color, texture, and spatial location. CompBench includes a diverse dataset curated from multiple publicly accessible sources, ensuring a wide range of domains and scenarios are covered.

---

> ### Author Rebuttal · Authors · 2024-08-17
>
> **Q. [Data bias]** Thank you for the insightful comment. We agree that it would be ideal to have a common data source for different types of comparative questions. However, in our humble opinion, it is difficult to find a single image domain that covers all the relativity types with sufficient diversity, difficulty, and amounts of data. For our current benchmark, we suggest that the users report accuracy per data source. We will seek to expand our benchmark to cover more data sources to reduce bias (for example, going beyond soccer to consider other dynamic events).
>
> Specifically for our current benchmark, we indeed implemented strategies to balance the data “within” each data source. For example, we set a cap of questions for each cluster of image pairs (e.g., a specific attribute in Line 155) to prevent dominating questions. We also balance the answers per question to prevent dominating answers (e.g., balance the number of questions with the answers “left” and “right”). We apologize that we missed these details in the current submission and we will clarify them in the final version.
>
> **Q. [Question Generation]** Thanks for your comment. We realize that we did not specify some details of our question-generation process in the current submission and we apologize for it. We indeed leveraged advanced MLLMs combined with a human-in-the-loop approach to generate and validate our questions. For instance, we utilized GPT4V with in-context learning to create questions for certain relativities, such as Existence (Lines 168-169). Additionally, human annotators reviewed the quality of the generated questions and made modifications when necessary (Figure 2). We will clarify this with additional descriptions in the final version. In the future, we will seek to apply LLMs to increase the question diversity for those currently based on rules (i.e., Sect. 4.2.1. and 4.2.5).
>
> **Q. [Limited Baselines]** Thank you for your comment. While we only included four MLLMs in the paper, in our humble opinion, they were either the strongest models or most accessible models at the time of submission. During the rebuttal, we add two more MLLMs (GPT4o, Gemini-1.5-Pro). As shown, the latest versions of GPT4 and Gemini have notable improvements in their comparative capability, while still worse than humans.
>
> |                  |            |            |            |            |            |              |              |      |      |                          |                          |      |              |              |              |      |      |
> |------------------|------------|------------|------------|------------|------------|--------------|--------------|------|------|--------------------------|--------------------------|------|--------------|--------------|--------------|------|------|
> |                  | **Attribute**  | **Attribute**  | **Attribute**  | **Attribute**  | **Attribute**  | **Exist.**   | **Exist.**   | **State** | **State** | **Emot.**                 | **Emot.**                 | **Temp.** | **Temp.**    | **Spat.**    | **Quan.**    | **Qual.** | **Avg** |
> | **Model**        | **ST**     | **FA**     | **VA**     | **CU**     | **WF**     | **MB**        | **SD**        | **ST** | **VA** | **CE**                   | **FE**                   | **SN** | **CC**        | **ND**        | **VQ**        | **QB** |      |
> | **GPT-4o**       | 92.3       | 97.0       | 86.3       | 74.7       | 84.5       | 81.2          | 67.2          | 95.8  | 89.6  | 96.6                     | 91.1                     | 72.0  | 83.3          | 68.2          | 67.8          | 81.2  | 83.1 |
> | **Gemini 1.5-Pro** | 79.2     | 91.8       | 77.7       | 71.4       | 72.8       | 55.4          | 58.7          | 91.0  | 84.0  | 93.0                     | 87.3                     | 50.3  | 70.3          | 68.3          | 64.8          | 70.5  | 74.2 |
>
> We will include these results in the final version and include a discussion.
>
> Regarding the fine-tuning experiment, we would like to reiterate that fine-tuning is not the main focus of our paper (Line 294 - 302). The need to access the model weights, not just the APIs, further shrinks the range of MLLMs we can consider. That said, your comment is well received, and we further include fine-tuning results with Qwen-VL-Chat [1]. For the quantity (VQAv2) questions, fine-tuning improves the accuracy from 44.5% (w/o fine-tuning) to 46.1%, which, however, is still far below human performance. This further supports our message in Lines 300 - 302: the comparative capability may not be improved simply through fine-tuning but may require new architectures or training strategies.
>
> References. [1] Bai et al. Qwen-VL: A Versatile Vision-Language Model for Understanding, Localization, Text Reading, and Beyond, arXiv, 2023.

---

> > ### Comment · Reviewer_MVH6 · 2024-08-30
> >
> > Thank you for your response. I'll raise the score

---

### Official Review · Reviewer_Z9Lw · 2024-07-17
**A Comparative Reasoning Benchmark for Multimodal LLMs**

**Rating:** 9
**Confidence:** 4
**Correctness:** NA
**Clarity:** yes

**Review:**

The paper provide a thorough benchmark on effectiveness of current MLLMs when it comes to comparatively reasoning. They provide a benchmark for future development; meanwhile they conduct series of analysis to leverage their data and assesss whether they can improve the performance of MLLMs.

**Strengths:**

introduce and point to an interesting common problem
they provide a data that can cover eight different aspects of comparative reseasoning
they set a baseline using prompt engineering but also fine-tuning
they conduct an error analysis

**Additional Feedback:**

1) line 302 : "Please see the supplementary material for further details." specify which section in the supplementary is relevant.
2) since the study focuses on visual comparison and MLLM is rapidly evolving field, I suggest choosing a name that specify the domain.

**Documentation:**

yes

**Limitations:**

- Possibility one of the limitation is on the size of the data - a few thousand per task does not look a lot - in the supplementary they also acknowledge the same issue.

- only looking into visual aspects

**Opportunities For Improvement:**

NA

**Relation To Prior Work:**

yes

**Summary And Contributions:**

the paper points to one of the MLLMs weakness; comparing pictures and answering factual information accordingly (they call it comparatively reasoning). They also introduce a comprehensive datasets to assess and cover 7 aspects of "relativity" types.

---

> ### Author Rebuttal · Authors · 2024-08-17
>
> **Q. [Data size]** Thank you for your comment. We plan to continue scaling up our benchmark by incorporating additional data sources for each relativity task, to increase both diversity and quantity. In our humble opinion, our annotation protocols can be conceptually expanded to many existing vision datasets. We will also conduct more human evaluations, as mentioned in the limitation in the supplementary.
>
> **Q. [Naming & Only visual aspects]** Thank you for your comment and suggestion. We will consider adding “visual” to our title. We would like to emphasize that, while our benchmark mainly focuses on the comparison in the visual aspects — where a test example involves a pair of images and a corresponding text question — solving the task requires MLLMs to have capabilities beyond the visual. For instance, to answer the question “Which frame happened first?” given two consecutive frames from the same soccer action (e.g., a corner kick), the model needs not only visual knowledge but also an understanding of soccer. Similarly, if both lemons are peeled and the question asks, “Which lemon is more peeled?”, the model needs to apply commonsense knowledge (i.e., understanding the degree of peeling) to answer correctly. That is, while we compare two images, the model should integrate various forms of knowledge beyond the visual to provide accurate answers.
>
> **Q. [Line 302]** Thank you for your feedback. Line 302 refers to Appendix C, and we will make this clearer.

---

### Official Review · Reviewer_V7ua · 2024-07-24
**CompBench Review**

**Rating:** 6
**Confidence:** 4
**Correctness:** Yes
**Clarity:** Yes

**Review:**

Strengths:
1. This benchmark paper addresses a critical gap in the evaluation of MLLMs which often struggle with tasks requiring fine-grained details, impacting their performance in answering comparative questions accurately.
2. The benchmark paper's data collection is comprehensive, covering a wide range of aspects such  Attributes, Existence, State, Emotion, Spatiality, and Quantity. This extensive coverage ensures that the evaluation captures various dimensions of MLLMs' performance, providing a detailed assessment of their capabilities across different types of tasks.

Weakness:

1.  While the paper provides an analysis on the inefficacy of two-stage reasoning compared to direct comparison, it would be interesting to explore the use of common strategies like chain of thought or providing MLLMs with more hints. How do these techniques impact the performance of MLLMs in such difficult tasks? An analysis of such strategies could provide valuable insights into their effectiveness and potential improvements.

2. In Table 4, the human performance is not as good as expected. I think that some comparative questions in the benchmark might be subjective, potentially leading to variability in responses. Have you analyzed this issue in your study? It would be helpful to understand whether subjective interpretations have been accounted for and if any data analysis has been conducted to measure their impact.

3. Given that this benchmark covers a wide range of aspects and requires various abilities from MLLMs, there is a concern that the data might not be sufficient to establish a robust learning-based baseline.

**Strengths:**

1. The submission provides relevant context for practical applications and ongoing research.
2. The submission employs a clear methodology for data collection and analysis, contributing to a thorough and well-organized evaluation of model performance.
3. The submission discuss the societal impacts.

**Additional Feedback:**

At this time, I do not have any additional feedback.

**Documentation:**

Yes.

**Limitations:**

The authors have made significant efforts to address the limitations and potential negative societal impacts of their work.

**Opportunities For Improvement:**

See Weakness in the Review.

**Relation To Prior Work:**

Yes

**Summary And Contributions:**

COMPBENCH aims to assess the  ability of multimodal large language models (MLLMs) to do comparative reasoning when given a pair of images.  It collects image pairs with visually oriented questions that cover eight dimensions of comparison: visual attribute, existence, state, emotion, temporality, spatiality, quantity, and quality. The whole dataset has approximately 40,000 image pairs from various visual domains like animals, fashion, sports, and scenes, both outdoor and indoor. This paper uses CLIP similarity scores for coarse data generation and then the human annotators label the questions for accuracy and relevance. For baselines, it evaluates GPT-4V(ision), Gemini-Pro, and LLaVA-1.6, revealing significant shortcomings in their comparative abilities.

---

> ### Author Rebuttal · Authors · 2024-08-17
>
> **Q. [Common strategies]** Thank you for the insightful comment. We want to reiterate that the purpose of our “two-stage reasoning vs. direct comparison” is to justify the necessary capability for an MLLM to directly look at and compare two images: the two-stage reasoning in our paper means the MLLM does not directly compare the two images visually (Line 284 - 293). In our humble opinion, common strategies like the Chain of Thought (CoT) or providing MLLMs with more hints can be added to both of them to further improve their performance. During the rebuttal, we followed your suggestion to explore these strategies for the direct comparison method. Concretely, given the two images *together*, we asked GPT4-V to first output descriptions of each of the two images as additional hints for answering the comparative question. This approach led to a 9.8% gain in GPT-4V’s performance on spatial relativity (i.e., GPT-4V with hints: 65.9% vs. GPT-4V: 56.1%). Although there is still a noticeable gap for further improvement, the enhanced results indicate that certain common strategies can be effective for comparative reasoning.
>
>
>
> **Q. [Human performance]** This is an excellent question. We are indeed aware of the potential subjectivity and made concerted efforts to address it while constructing our benchmark. Specifically, we implemented a rigorous cross-verification process where each annotator confirmed the accuracy of others’ answers. Only samples that received unanimous approval from the annotators were retained in our benchmark (Lines 224-226). We would be happy to extend our analysis of subjective interpretations, for example, by conducting a qualitative review of cases where human responses varied.
>
> We thus attribute the non-exceptional human performance (86.5%) to the inherent difficulty of our questions rather than their subjectivity. Some of our questions are challenging to answer correctly at first glance, even when there exists a clear objective answer. For instance, a question like “Which image has more elephants?” can be difficult if the elephants are hard to recognize in the image, such as when they are small or positioned far away. In such cases, evaluators may struggle to answer correctly without thoroughly examining the image. In other words, we think the performance depends more on how carefully the evaluators answer our questions, rather than on subjective interpretation.
>
>
> **Q. [Data insufficiency]** Thanks for your concern. We would like to emphasize that the main goal of our benchmark is to evaluate existing MLLMs (that have been pre-trained with a significant amount of data), rather than to be used for training MLLMs. This follows the recent trends of evaluation benchmarks in the MLLM community.
>
> That said, we believe that our annotation protocol can be further utilized to generate sufficient data to establish a robust learning-based baseline. For instance, several data sources used in our benchmark (e.g., MagicBrush) contain training sets, though we only used their test sets. By applying our annotation protocols to these training sets, we can generate enough comparative samples for model training. As an example, in this submission, we demonstrate this with two datasets, SoccerNet and VQAv2 (Lines 155-160 in the appendix). We showed that turning their training data into training examples for comparative questions could be used to fine-tune MLLMs effectively (Table 3, Right).

---

> > ### Comment · Reviewer_V7ua · 2024-08-29
> >
> > Happy to see the results of using Chain of Thought (CoT) and thanks for the explanations provided for the other concerns. I will maintain my score.

---

### Author Response · Authors · 2024-08-17
**General Response**

We thank the reviewers for their valuable comments.
They found our benchmark “addressing a critical gap in the evaluation of MLLMs” (V7ua, MVH6), and “a valuable resource for advancing MLLM development” (MVH6); our data collection “comprehensive” (V7ua, Z9Lw, MVH6, xgQo); our analysis “detailed and well-organized” (xgQo, V7ua); our idea “innovative” (MVH6); our problem “interesting” (Z9Lw). We address major comments below and will incorporate all your feedback in the final version.

---

### Decision · Program_Chairs · 2024-09-26

**Decision:**

Accept (Poster)

**Comment:**

This paper points to one of the MLLMs weakness and contributes a new benchmark to evaluate the comparative reasoning capabilities of MLLMs. The introduced benchmark consists of around 40K image pairs across various visual domains, each paired with a question and answer that focus on eight dimensions of relative comparison. This paper also evaluates several recent MLLMs and find significant shortcomings in comparative abilities of MLLMs, particularly in existence, spatiality, and quantity relativity. This paper also provides a detailed analysis of error cases and insights for future improvements.

All reviewers agreed to accept this paper and find the following strengths:
- Addressing a critical gap in the evaluation of MLLMs
- A valuable resource for advancing MLLM development
- A thorough analysis of model performances